# The development and persistence of soil moisture stress during drought across Southwestern Germany

Erik Tijdeman and Lucas Menzel

Institute of Geography, Professorship in Hydrology and Climatology, Heidelberg University, Heidelberg, Germany

*Correspondence to*: Erik Tijdeman (erik.tijdeman@uni-heidelberg.de)

**Abstract**

The drought of 2018 in Central and Northern Europe showed once more the large impact this natural hazard can have on the environment and society. Such droughts are often seen as slowly developing phenomena. However, root zone soil moisture deficits can rapidly develop during periods of lacking precipitation and meteorological conditions that favour high evapotranspiration rates. These periods of soil moisture stress can persist for as long as the meteorological drought conditions last, thereby negatively affecting vegetation and crop health. In this study, we aim to characterize past soil moisture stress events over the croplands of Southwestern Germany, and further to relate the characteristics of these past events to different soil and climate properties. We first simulated daily soil moisture over the period 1989-2018 on a 1-km resolution grid using the physical based hydrological model TRAIN. We then derived various soil moisture stress characteristics; probability, development time and persistence, from the simulated time series of all agricultural grid cells (n ≈ 15000). Logistic regression and correlation were then applied to relate the derived characteristics to the plant-available storage capacity of the root zone as well as to the climatological setting. Finally, sensitivity analyses were carried out to investigate how results changed when using a different parameterization of the root zone, i.e., soil based or fixed, or when assessing soil moisture drought (anomaly) instead of stress. Results reveal that the majority of agricultural grid cells across the study region reached soil moisture stress during prominent drought years. The development time of these soil moisture stress events varied substantially, from as little as 10 days to over 4 months. The persistence of soil moisture stress varied as well and was especially high for the drought of 2018. A strong control on the probability and development time of soil moisture stress was found to be the storage capacity of the root zone, whereas the persistence was not strongly linearly related to any of the considered controls. On the other hand, the sensitivity analyses revealed the increased control of climate on soil moisture stress characteristics when using a fixed instead of a soil-based root zone storage. Thus, the strength of different controls depends on the made modelling assumptions. Nonetheless, storage capacity of the root zone, whether it is a characteristic of the soil or a difference between a shallow or deep rooting crop, remains an important control on soil moisture stress characteristics. This is different for SM drought characteristics, which show little or a contrasting relation with the storage capacity of the root zone. Overall, results give insights in the large spatial and temporal variability of soil moisture stress characteristics and suggest the importance of considering differences in root zone soil storage for agricultural drought assessments.

# 1 Introduction

Droughts are naturally (re-)occurring phenomena that can appear in different domains of the hydrological cycle and cause associated impacts (Tallaksen and Van Lanen 2004; Stahl et al., 2016). Because of their multifaceted characteristics, droughts are often classified in different types (Wilhite & Glantz, 1985). One of these drought types is agricultural drought, which refers to the impacts of lacking water availability on the health and growth of crops. These agricultural droughts can reduce yields and thereby cause large economic losses. A crucial first step to reduce the risk of (agricultural) drought impacts involves effective monitoring and early warning of the drought hazard (UN/ISDR, 2009). Agricultural drought monitoring and early warning occurs at different scales; from plot-scale observations and simulations to regional-scale drought mapping. Regional-scale drought monitoring and early warning provides an overview of regions at drought risk, which raises awareness and helps decision-making. Accurately depicting areas affected by agricultural drought is complex as its occurrence is influenced by a variety of factors, including often spatially heterogeneous climate and soil characteristics. A better understanding how these climate and soil characteristics control (the development of) agricultural droughts is needed.

Droughts are often defined as a below normal water availability, with the normal often depending on space and time (Tallaksen and Van Lanen 2004). Such an anomaly-based definition allows depicting regions and episodes with below normal water availability across the world according to different hydro-meteorological variables. However, the identified events with below normal water-availability might not necessarily have the potential to cause drought related impacts. The below-normal definition of drought forms the basis of many drought indices, which reflect whether a certain hydro-meteorological variable is anomalously low or high (e.g., Anderson et al., 2007; McKee et al., 1993; Samaniego et al., 2012; Vicente-Serrano et al., 2010). Soil moisture anomaly time series, or proxies of the latter, are often used for agricultural drought assessments (e.g., Sheffield et al., 2004; Andreadis et al., 2005; Samaniego et al., 2012). Different drought characteristics can be derived from these soil moisture anomaly time series, including drought magnitude, duration, and areal extent.

The data used for agricultural drought assessments stems from different sources. These data sources include direct soil moisture measurements, remote sensing observations, meteorological proxies and hydrological- or land surface model simulations (e.g., Berg and Sheffield, 2018). Soil moisture measurements provide the most realistic information about the soil moisture status at a certain depth but are point based and thereby limited in their spatial coverage. Remote sensing observations of soil moisture provide a regional coverage but direct observations are only able to detect soil moisture changes in the upper soil layer, at least in the case of microwave remote sensing. On the other hand, remote sensing observations of heat fluxes and vegetation health can provide an estimate of the ratio between actual and potential evapotranspiration and thereby depict regions with soil moisture stress (e.g. Anderson et al., 2007). Meteorological proxies for agricultural drought include drought indices such as the Palmer Drought Severity Index (PDSI; Palmer, 1965) or Standardized Precipitation Evapotranspiration Index (SPEI, Vicente-Serrano et al., 2010). The strength of these meteorological proxies is their relative ease of computation and often low data requirements. However, meteorological proxies are often based on potential evapotranspiration and do not consider some other relevant terrestrial processes that influence soil moisture and agricultural drought, such as the reduction of

evapotranspiration during soil moisture stress. Many of these terrestrial processes are also included in physical-based hydrological and land surface models. The physical basis of these models makes their use often preferable over the use of meteorological proxies for past and future agricultural drought assessments (e.g., Berg & Sheffield, 2018; Sheffield et al., 2012).

Various hydrological and land surface models have been used to assess past and future soil moisture drought events. One example is the Variable Infiltration Capacity model (VIC), which has been applied to characterize major soil moisture drought episodes across different regions (e.g., US: Sheffield et al., 2004; Andreadis et al., 2005; China: Wang et al., 2011: and the world: Sheffield & Wood, 2007). The latter analyses enabled the cataloguing of past soil moisture drought events according to a variety of characteristics, providing a benchmark for current and future drought events. Another example of a regionally

applied model to simulate soil moisture (drought) is the mesoscale Hydrological Model (mHM, Samaniego et al., 2010). The output of the mHM has been used for both historic soil moisture drought assessments (Hanel et al., 2018) as well as for future soil moisture drought projections according to different climate change scenarios across Europe (as part of a model ensemble in Samaniego et al., 2018). The latter studies provide valuable insights about the severity of recent soil moisture drought events over Europe, e.g., 2003 and 2015, and also show that these recent events were not as rare when considered in a more long-

term historical perspective and that similar or worse events are likely to occur under different climate change scenarios. The mHM is also run in near-real time and its output is used by the German Drought Monitor (Zink et al., 2016).

Studies mentioned in the previous paragraph focus on characterizing past and future soil moisture drought events, whereas other studies aim to characterize its development. Drought is often referred to as a slowly developing phenomena, that can take up to years to reach its full extent (Wilhite & Glantz, 1985). However, not all drought events are slowly developing

phenomena and soil moisture deficits can develop relatively quickly during dry weather conditions that favor high amounts of evapotranspiration (e.g., Hunt et al. 2009). These rapid developing droughts, sometimes termed "flash droughts", can severely impact agriculture (e.g., Svoboda et al., 2002, Otkin et al., 2018). Several case-study flash drought events in the US have been described in Otkin et al. (2013; 2016). The latter studies show that precipitation deficits can be quickly followed by a reduction of evapotranspiration, which is indicative for low soil moisture levels causing water stress for plants. Christian et al. (2019)

aimed to make a regional assessment of past flash droughts and developed a framework of objective criteria to identify flash drought events from simulated soil moisture output. By applying this framework to soil moisture simulations over the US, they show that particular regions, such as the Great Plains, are more sensitive to flash drought occurrence.

Most of the above-described soil moisture drought assessments characterize drought as a below normal anomaly according to different hydrometeorological variables, which is in line with the traditional definition of drought. However, from an

agricultural drought impact perspective, it can make more sense to directly study the characteristics of (the development of) periods of lacking amounts of root zone soil moisture, i.e., soil moisture stress, which is in line with the soil moisture drought index proposed in Hunt et al. 2009. Following this reasoning and inspired by the methods used in previous soil moisture anomaly studies, we aim to study simulated soil moisture stress events across the agricultural regions of Southwestern Germany. Our objectives are to:

1) Characterize past soil moisture stress events,

2) Investigate dominant controls on soil moisture stress characteristics

3) Portray meteorological anomalies during (the development of) soil moisture stress

Finally, we aim to carry out a sensitivity analyses to investigate how derived (controls on) characteristics change when using different parametrizations of the root zone soil or when investigating soil moisture drought instead of soil moisture stress.

## 2 Data and methods

### 2.1 Study region

The study region encompasses Baden-Württemberg (area $\approx 36000$ km$^2$), a federal state of Germany located in the Southwestern
part of the country (Fig. 1). The area of interest covers both flat and lowland regions such as the Rhine valley as well as higher located, more mountainous regions such as the Black Forest and the Swabian Jura (Fig. 1a). The topography of the study region affects both temperature (annual average ($T_{annual}$) between 4.5 °C and 11.6 °C, Fig. 1b) and precipitation (annual average sum ($P_{annual}$) between < 600 mm and > 2000 mm, Fig. 1c). Land cover and soil characteristics vary over the study region (Fig. 1d,e). Most of the  cropland, on which this study focusses, is located in the lower areas (Fig. 1d). Thicker soils with a higher available
water-holding capacity (AWC in mm, i.e.,  the amount of plant available water in the root zone at field capacity) are generally found in the valleys, and more shallow soils with a lower AWC in the higher elevated, mostly forested regions (Fig. 1d, e).

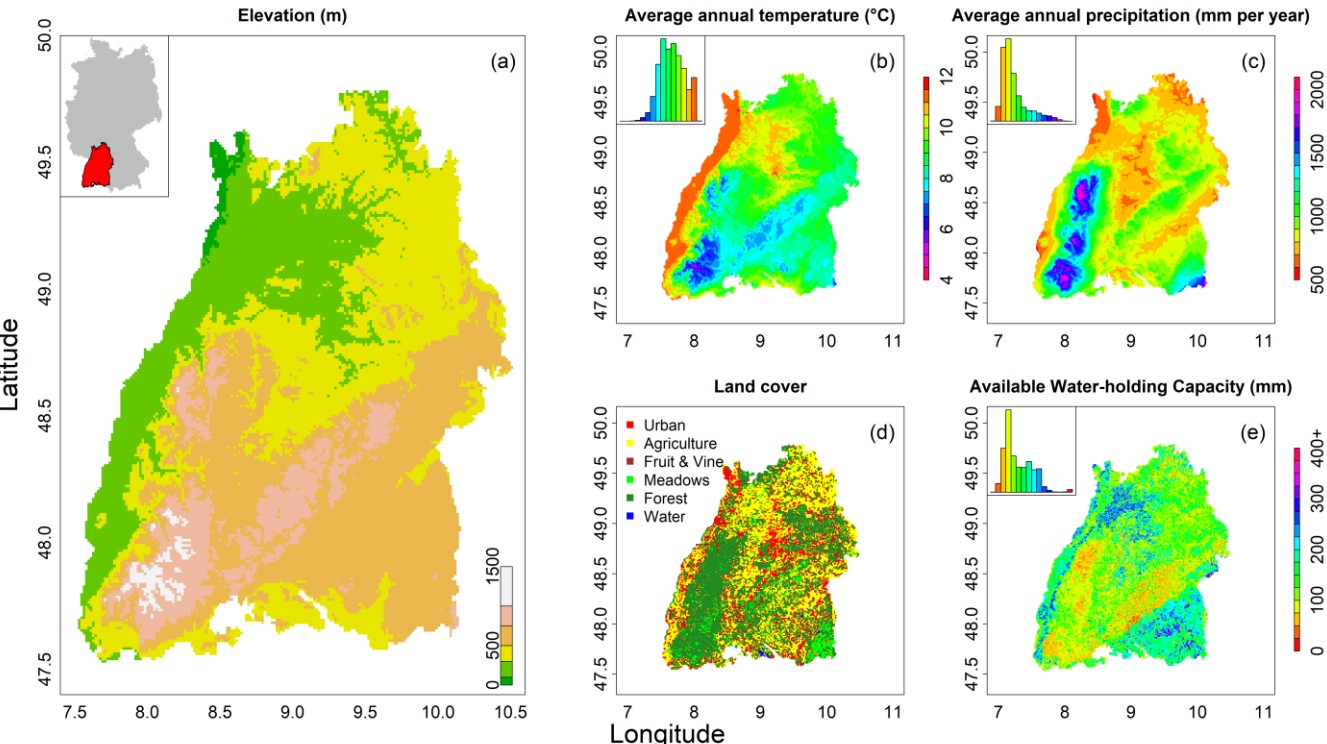

**Figure 1: Study region and its (a) elevation, (b) average annual temperature, (c) average annual precipitation sum, (d) land cover and (e) available water-holding capacity of the root zone soil. Gridded data used to derive this Figure are described in Section 2.2.**

### 2.2 Data and interpolation

The data used in this study stem from various sources. Gridded elevation data (1-km resolution) were obtained from the Federal Agency for Cartography and Geodesy (BKG, 2019). Vectorized land cover data come from the Corine-2006 dataset and were retrieved from the German Environment Agency (UBA, 2018). Vectorized soil property data (field capacity, wilting point, air capacity and depth of the root zone soil based on soil properties of different layers) were derived from the BK-50 (scale of 1:50,000) dataset provided by the Federal State Office for Geology Resources and Mining (LGRB, 2019). River flow data comes from the Environment Agency of Baden-Württemberg (LUBW). Daily meteorological data for the period between 1989-2018 used in this study stem from both gridded data as well as station-based observations. Gridded precipitation ($P$, mm) comes from the REGNIE dataset (Rauthe et al., 2013) and was sourced from the climate data center of the German Weather Service (DWD, 2019). Gridded satellite based global radiation data (W m$^{-2}$) stem from the SARAH dataset and were derived from the Satellite Application Facility on Climate Monitoring (Pfeifroth et al., 2019a,b). Station-based meteorological observations of temperature ($T$, °C), relative humidity (%) and sunshine duration (hours) as well as sub-daily observations of wind speed (Bft) and wind direction (degrees °) originate from the climate data center of the German Weather Service (DWD, 2019). The sub-daily values of wind speed and wind direction were aggregated to daily values (for wind speed: arithmetic average, for wind direction: average of Cartesian coordinates).

All data were interpolated to 1-km resolution grids covering Baden-Württemberg. Land cover and soil property data were interpolated based on the majority class within each grid cell. Gridded meteorological data were re-projected to match the extent and resolution of the soil and land cover grids. Station-based meteorological observations were interpolated to grids using the INTERMET software (Dobler et al. 2004; software ran in default settings). The software first converts (the units of) some of the meteorological observations, i.e., wind speed (Bft) to wind speed (m s$^{-1}$) and sunshine duration to global radiation. The software then interpolates these (and all other) meteorological observations to daily grids using different kriging-based interpolation techniques. These interpolation techniques consider distance to the station, and, depending on the variable, the possible relationship between the variable of interest and other external factors such as elevation, wind direction, or relief. The grids of global radiation interpolated with INTERMET were only used for days for which the SARAH dataset did not provide any data (< 0.25 % of days).

## 2.3 Soil moisture modelling

We applied the physically based hydrological model TRAIN (TRAnspiration and INterception, indicating the major processes considered during the initial phase of model development; Fig. 2). The model was used to simulate different fluxes, such as the different components of evapotranspiration ($E_{total}$) and percolation ($Q_{percolation}$) as well as stores such as soil moisture (SM) at a daily resolution over Baden-Württemberg. The TRAIN model follows some basic principles, of which the most important ones are the applicability of the model on both the plot and the areal scale (e.g., Stork & Menzel, 2016; Törnros & Menzel, 2014) as well as the ability to run the model with as few input data as possible, which benefits its general applicability on larger scales. TRAIN includes information from comprehensive field studies of the water and energy balance for different surface types, including natural vegetation and cropland (Menzel, 1997; Stork & Menzel, 2016). Special focus in the model is on the water and energy fluxes at the soil-vegetation-atmosphere interface.

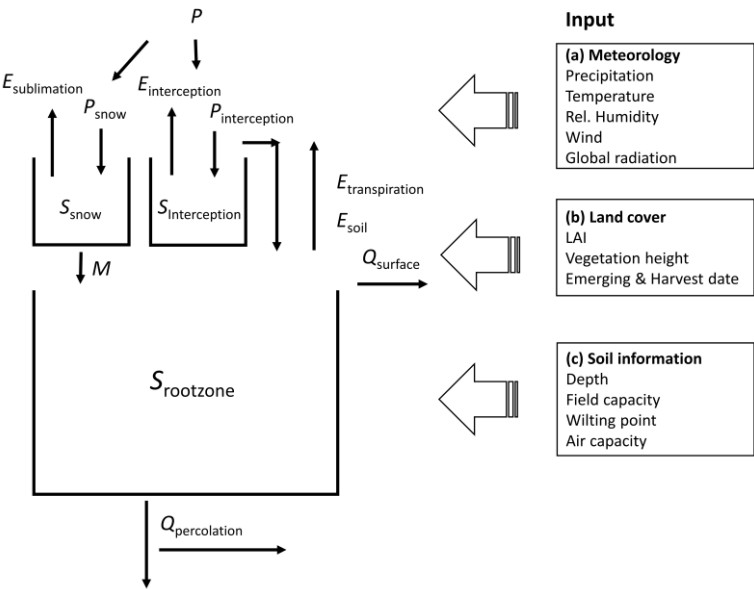

**Input**

**(a) Meteorology**
Precipitation
Temperature
Rel. Humidity
Wind
Global radiation

**(b) Land cover**
LAI
Vegetation height
Emerging & Harvest date

**(c) Soil information**
Depth
Field capacity
Wilting point
Air capacity

**Figure 2. Conceptual flowchart of the fluxes and stores considered in TRAIN.**

In brief, the model works as follows. First, precipitation is divided in either rain ($P_{rain}$) or snow ($P_{snow}$), depending on whether daily average $T$ exceeds the threshold temperature ($T_{threshold} = 0$ °C) or not. $P_{snow}$ is temporary accumulated in a snow storage reservoir ($S_{snow}$), which grows via accumulation of $P_{snow}$ or shrinks via melt ($M$, occurring when $T > T_{threshold}$ and derived using the degree day method; Kustas et al., 1994) or sublimation ($E_{sublimation}$; derived following the Penman-Monteith equation, with canopy resistances set to zero; Wimmer et al., 2009). $P_{rain}$ is either stored as interception ($S_{interception}$), where the size of $S_{interception}$ depends on the leave area index (LAI), or bypasses the interception reservoir if it is (partly) filled or non-existent. Water is removed from the interception reservoir via evaporation ($E_{interception}$), which is modelled to occur at different intensities as a function of the $S_{interception}$ and the present meteorological conditions (Menzel, 1997). $P_{rain}$ and $M$ either infiltrate in the root zone storage reservoir ($S_{rootzone}$) or generate surface runoff ($Q_{surface}$). The total water storage capacity of $S_{rootzone}$ is divided in different parts, i.e., immobile water (the volume of water below wilting point) plant available water (the volume of water between permanent wilting point and field capacity; also referred to as AWC) and excess water (volume of water above field capacity; constrained by the total porosity of the root zone soil). $Q_{surface}$ is only generated when $S_{rootzone}$ is saturated and $P$ exceeds an intensity threshold of 20 mm day$^{-1}$. The simulation of transpiration ($E_{transpiration}$) is based on the Penman-Monteith equation. It depends on the calculation of canopy resistances, which are modified by the state of growth of the vegetation, the status of $S_{rootzone}$ and the meteorological conditions (Menzel, 1996, box (a) of Fig. 2). The calculation of percolation ($Q_{percolation}$) follows the conceptual approach from the HBV-model (Bergström, 1995) and occurs at a rate that is a function of the amount of excess water in the root zone. Land cover properties are related to vegetation development, i.e., the temporal dynamics of LAI and vegetation height as well as emerging and harvest date in case of agriculture (Fig. 2, box b). They were derived from the Corine

dataset (Section 2.2), which encompass general land use classes, such as broadleaved forest or agriculture. Each of these land use classes were assigned associated temporally varying vegetation properties that are typical for the study region. For the agricultural grid cells, on which we focus in this study, we considered a mixed parameterization of typical agricultural crops of the region. It should be noted that in reality, there are crop specific differences that further vary in space and time due to e.g. spatiotemporal differences in climate or temporal changes in climate or genotypes (e.g. Bohm et al., 2019; Ingwersen et al., 2018; Rezaei et al., 2018). However, given the absence of detailed spatiotemporal information over the region about these differences, we used the generalization as described above.

$S_{\text{rootzone}}$ was derived from soil properties from the BK-50 dataset (Section 2.2, Fig. 2 box c). This dataset is based on extensive field investigations on soil profiles distributed over the whole of Germany, which led to a detailed soil map, including information about soil types, grain size distribution, sequence and depth of soil horizons as well as parameters describing the water-holding capacity (field capacity, wilting point, air potential). In addition, it includes information about the potential depth of the root zone, broadly ranging between a few decimeters up to two meters and constraint by e.g. the occurrence of a root restrictive layer. In addition to soil properties, other factors, such as plant type, climate and meteorological conditions during certain growth stages influences how deep plant roots grow and thereby the AWC of the root zone (e.g. Fan et al., 2016; de Boer-Euser et al., 2016). However, we used the above-described soil-based parametrization of $S_{\text{rootzone}}$ (more commonly used in regional modeling studies), as detailed spatiotemporal information about these other factors are unknown, and the used soil-based parameterization provides a reasonable boundary condition.

The initial conditions of $S_{\text{rootzone}}$ were set to field capacity at the start of the model run on the first of January of 1988. The first year (1988) was used as warm-up year, whereas the following 30 (1989-2018) years were used for the analyses. A longer warm-up was not needed for the purpose of this study given that only the amount of snow that accumulated in the winter of 1988-1989 affected the considered fluxes and stores over the studied period. Snapshots of the soil moisture status during different stages of the drought year 2018 are shown in Figure 3; complete daily animations of soil moisture status during different drought years are stored in an online repository (Tijdeman and Menzel, in review). This online repository also contains all daily simulations of soil moisture.

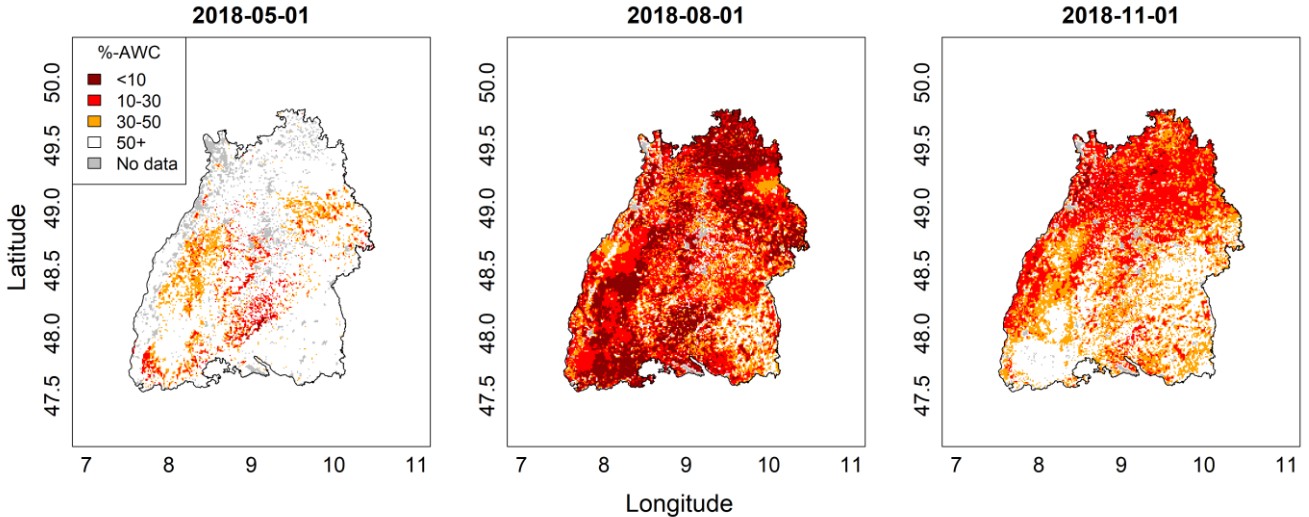


**Figure 3. Simulated soil moisture (expressed as the % of AWC left in the root zone) during different stages of the drought of 2018.**

In this study, we specifically analyzed simulated soil moisture (SM, expressed as the % of AWC left in the root zone) and simulated total evapotranspiration ($E_{total}$, mm day$^{-1}$). For the SM stress analyses, we focus on grid cells classified as agricultural, as the focus of this study is on agricultural drought. Grid cells of other land uses were only considered for the model evaluation.

**2.4 Model evaluation**

On the plot scale, the performance of TRAIN was evaluated against observed SM and $E$ observations in various previous studies (e.g. Sect. 2.3). However, such observations are scarcely available on the regional scale. Therefore, evaluation of the simulated fluxes and states vs. observed streamflow is helpful to get insight in whether these are reasonable or not. Given that TRAIN is not a rainfall runoff model a direct comparison between daily streamflow simulations and observations ($Q_{observed}$) is

not possible (Fig. 2). Rather, we evaluated for 60 catchments with near-natural flow located across the study region (Fig. S1), whether:

    1)   The average annual water balance is comparable ($Q_{percolation} + Q_{runoff} + \Delta S \approx Q_{observed}$); where $\Delta S$ is the change in catchment storage over the period of record.

2)   The annual water balance is comparable and correlated.

    3)   Monthly sums of $Q_{observed}$ are correlated with the sum of $Q_{percolation}$ and $Q_{runoff}$ accumulated over a catchment specific time window of n-months

    4)   The gradual drying of simulated SM during meteorological drought is also visible for part of the $Q_{observed}$ timeseries, i.e., those without a large groundwater flow contribution that can sustain low flows.

5) Event- or quick-flow mainly occurs when simulated SM exceeds field capacity for most grid cells within the catchment.

Several storage components encompassed in $\Delta S$, e.g. $S_{snow}$ or $S_{rootzone}$, are simulated in the TRAIN model; however, groundwater is not (Fig. 2). For the first criterion, the impact of not considering groundwater in $\Delta S$ is relatively small as the sums of $P$, $E_{total}$ and $Q_{observed}$ over the considered period are much larger. For the second and fourth criteria, not considering

groundwater in $\Delta S$ can have a larger influence, especially for catchments with extensive groundwater stores that can buffer low flows even though $S_{rootzone}$ is depleted. For the third criterion, we considered differences in catchment response using the following approach (inspired from Barker et al., 2016). We first accumulated the sum of $Q_{percolation}$ and $Q_{runoff}$ over n-month periods (1-12 months), i.e., for each month the sum of $Q_{percolation}$ and $Q_{runoff}$ in the current month, the current and previous month etc. (similar to the calculation of the SPI-n). Following, we correlated monthly $Q_{observed}$ with the sum of $Q_{percolation}$ and $Q_{runoff}$

accumulated over the different n-month periods for each catchment and calendar month. In the end, we selected for each catchment and calendar month the accumulation period with the maximum correlation with $Q_{observed}$.

## 2.5 Soil moisture stress characteristics

We identified SM stress events, i.e., events where SM was continuously at or below a threshold ($\tau$), from all daily simulated SM time series of agricultural grid cells. In this study, $\tau$ was set to 30% of the AWC (i.e., 30% of available water left in the

root zone), which is in line with the threshold used by the German Weather Service to define possible low-water stress (DWD, 2018). Various characteristics were calculated for the identified SM stress events. We first created a binary time series of annual SM stress occurrence ($S_{occ}$) for each agricultural grid cell (i = 1, 2 … 15359) and calendar year (y = 1989, 1990 … 2018), which indicates for each grid cell and each year whether SM stress was reached ($S_{occ,i,y} = 1$) or not ($S_{occ,i,y} = 0$). Then, if $S_{occ,i,y} = 1$, i.e., grid cell i reached SM stress in year y, various other SM stress characteristics were derived for that grid cell

and year, namely:

| | |
|---|---|
| $S_{start,i,y}$ | The first day of SM stress (doy) |
| $S_{devtime,i,y}$ | The development time of SM stress (days), i.e., the time it took to drop from field capacity (last day) to SM stress (first day). |
| $S_{total,i,y}$ | The total time in SM stress (days), i.e., the number of days $SM_{i,y} < \tau$ |
| $S_{maxdur,i,y}$ | The maximum duration of SM stress (days), i.e., the maximum number of consecutive days with $SM_{i,y} < \tau$ |

These different SM stress characteristics are exemplified in Figure 4. In this study, SM stress episodes were defined based on the percentage of water left in the soil. Thus, SM stress differs from SM drought, which is expressed as anomaly.

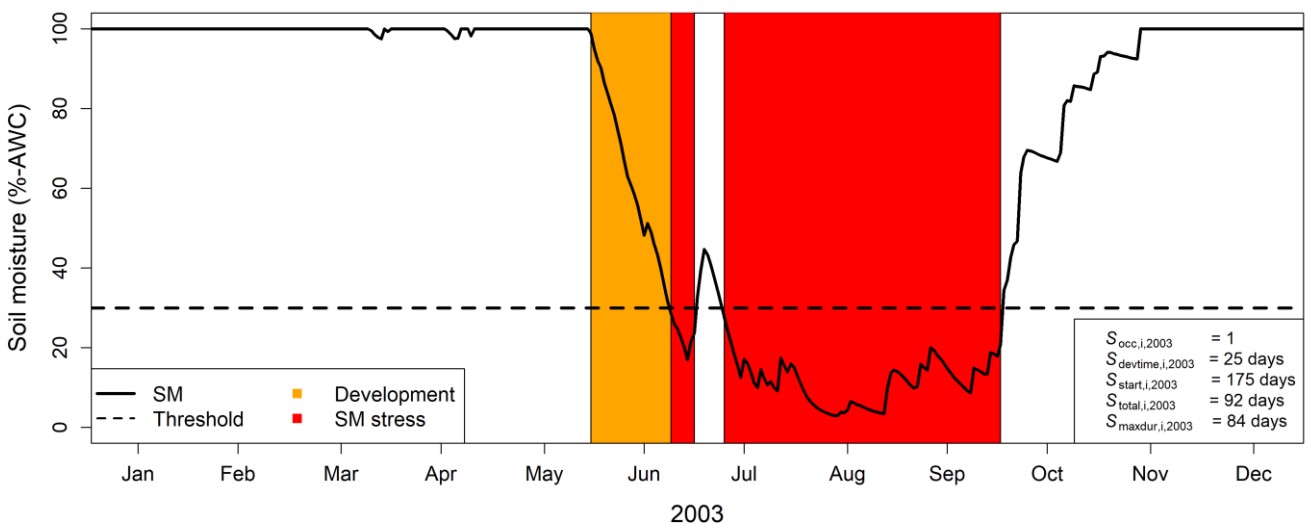


**Figure 4. Simulated soil moisture (SM) time series of an exemplary agricultural grid cell (i) showing the development and persistence of SM stress in 2003. The considered SM stress characteristics are presented in the lower-right legend, i.e., whether SM stress developed or not ($S_{occ,i,2003}$) and the development time ($S_{devtime,i,2003}$), first day ($S_{start,i,2003}$), total number of days ($S_{total,i,2003}$) and maximum duration ($S_{maxdur,i,2003}$). In this plot, SM time series are capped at 100%-AWC.**

**2.6 Controls on simulated SM stress characteristics**

We related the derived SM stress characteristics in different years (y) to the soil properties (AWC, Figure 1e) and climatological setting ($T_{annual}$ & $P_{annual}$, Figure 1b,c). Two different techniques were used:

1)      Logistic regression for the binary data of $S_{occ,y}$

2)      Spearman's Rank correlation for the integer time series of $S_{start,y}$, $S_{devtime,y}$, $S_{total,y}$ and $S_{maxdur,y}$

Both the logistic regression and correlation analyses were carried out for each year separately to investigate whether the results were consistent over the years or exhibit a year-to-year variability.

**2.7 Meteorological anomalies during (the development of) SM stress**

We further characterized the meteorological anomalies during (the development of) SM stress. For all grid cells and years (and when $S_{occ}$=1), we calculated anomalies of $P$, $T$, and $E_{total}$ (percentiles; resp. $P_{perc,i,y}$, $T_{perc,i,y}$, and $\underline{E}_{perc,i,y}$) during both the development (dev) and annual maximum duration (maxdur) of SM stress. Weibull plotting positions were used to calculate these percentiles, i.e., rank(x)/(n+1); where x is the meteorological variable of interest and n the sample size (in this study, n=30 years). The time window for which these percentiles were derived matches the time window of development and annual

maximum duration. For the example in Figure 4, SM stress developed between the 31st of May and 24th of June and had its

maximum duration between the 10[th] of July and 1[st] of October of 2003. For this event, $P_{perc,dev,i,2003}$, $T_{perc,dev,i,2003}$ and $E_{perc,dev,i,2003}$ ($P_{perc,maxdur,i,2003}$, $T_{perc,maxdur,i,2003}$ and $E_{perc,maxdur,i,2003}$) express the meteorological anomalies of the period between the 31[st] of May and the 24[th] June (10[th] of July and 1[st] of October) in 2003, relative to the same time window in all other years.

For ease of notation, we omit the grid cell identifiers (i) and where applicable year identifiers (y) from the variable subscripts in the remainder of this paper.

### 2.8 Sensitivity to the parameterization of $S_{rootzone}$ and used identification method

The AWC of $S_{rootzone}$ was derived from properties of the root zone soil (Sect. 2.3; from now on referred to as soil based $S_{rootzone}$). To investigate the sensitivity of the derived (controls on) simulated SM stress characteristics to the parameterization of the $S_{rootzone}$, we carried out the same analyses but with simulations derived using different root zone parameterizations. For one parameterization, the AWC of the root zone was again based on soil properties but the depth of the root zone soil was constraint at one meter, placing a fixed lower boundary on rooting depth. For two other parameterizations, we fixed the size of the AWC of $S_{rootzone}$ to resp. 100 mm and 200 mm, aiming to differentiate between (more shallow rooting) crops with a lower water availability and (deeper rooting) crops with a higher water availability.

SM stress episodes were defined based on the percentage of plant-available water left in the root zone soil. However, given that percentage of water left in the soil differs from SM anomalies commonly used for drought studies, we carry out a sensitivity analyses to investigate how (controls on) SM stress characteristics differ from (controls on) SM anomaly characteristics, hereafter referred to as SM drought. For this comparison, daily SM values were first transferred to anomaly space using Weibull plotting positions (Sect. 2.7), ranking daily SM values of a certain calendar day and year compared to SM values of the same calendar day in other years. The 20[th] percentile threshold commonly used for drought studies was used to extract drought episodes from the SM anomaly time series. Then, (controls on) the characteristics of these drought episodes were characterized in the same way as was done for SM stress episodes (Sect. 2.5, 2.6).

## 3 Results

### 3.1 Model evaluation

Overall, annual average $Q_{observed}$ reveals a good agreement with the sum of annual average simulated $Q_{percolation}$ and $Q_{runoff}$ (Fig. 5a). Differences are mostly within the 100 mm range, with few exceptional catchments showing slightly larger differences, especially in the wetter domains of the study region encompassing mostly forested catchments. Systematic biases related to the catchment average AWC of the root zone were not observed. Figure 5b reveals the distribution of Spearman's rank correlation coefficients between annual $Q_{observed}$ and the simulated annual sum of $Q_{percolation}$ and $Q_{runoff}$ (averages over the hydrological year) for all catchments. The generally high correlation coefficients indicate that TRAIN simulates the inter-annual variability more or less right, especially when considering that TRAIN does not have a base flow reservoir and therefore is not able to simulate (annual) variability in groundwater storage. On the monthly scale, the correlation between $Q_{observed}$ and

the sum of $Q_{percolation}$ and $Q_{runoff}$ accumulated over the n-month period with the highest correlation with $Q_{observed}$ indicated a good agreement (Fig. S2). Further, their percentile time series were comparable during prominent drought years 2003 and 2018 (Fig. S3a-d). In addition, episodes with anomalously low SM generally coincide with episodes of anomalously low river

flow as is exemplified for drought years 2003 and 2018 in Fig. S3. Finally, Figure S4 and S5 reveal that a relatively large proportion of precipitation contributes to event flow whenever $S_{rootzone}$ of all grid cells within the catchment are filled to a level at or above field capacity. This relative contribution of precipitation to event flow strongly declines whenever a large proportion of grid cells within the catchment drop to a level below field capacity.

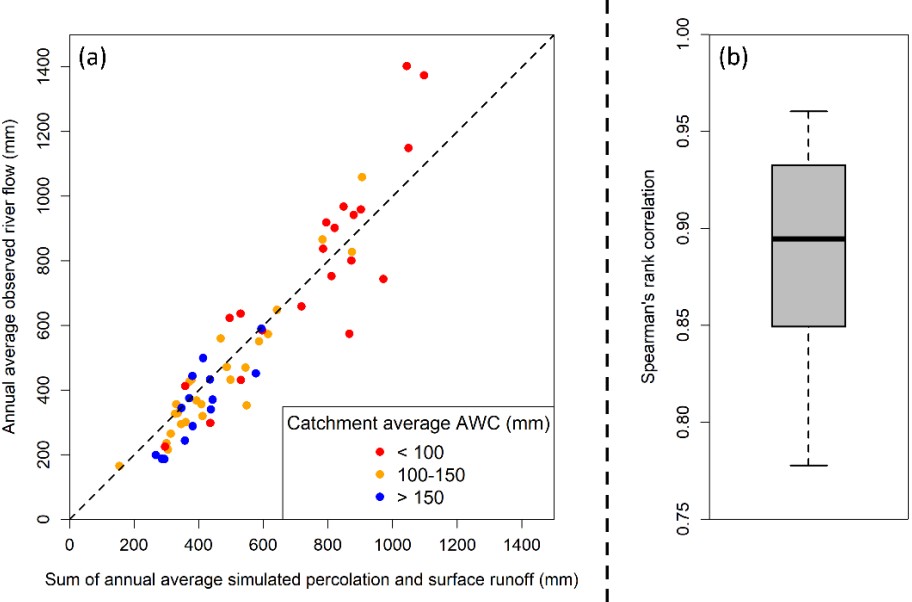


**Figure 5. (a) Annual average $Q_{observed}$ vs. the annual average sum of simulated $Q_{percolation}$ and $Q_{runoff}$ (each dot reflects one catchment; colour of the dots indicates catchment average AWC; dashed red line is the 1:1 line) and (b) distribution of Spearman's rank correlation between annual $Q_{observed}$ and the sum of simulated annual $Q_{percolation}$ and $Q_{runoff}$ considering hydrological years (October - September) for all considered catchments. Box: percentiles 25, 50 and 75. End of whiskers: percentiles 5 and 95.**

**3.2 (Controls on) past SM stress characteristics**

Figure 6 presents the percentage of grid cells that reached SM stress at least once in different calendar years ($S_{occ} = 1$). In general, results reveal a large temporal variability in the fraction of cells that reached SM stress. SM stress was reached in all years for at least a small proportion of the cells. However, most prominent drought years (i.e., the years in which most cells reached SM stress) were 2003 and 2018, followed by 2015 and 1991. During these years, up to 89% of the grid cells reached

SM stress.

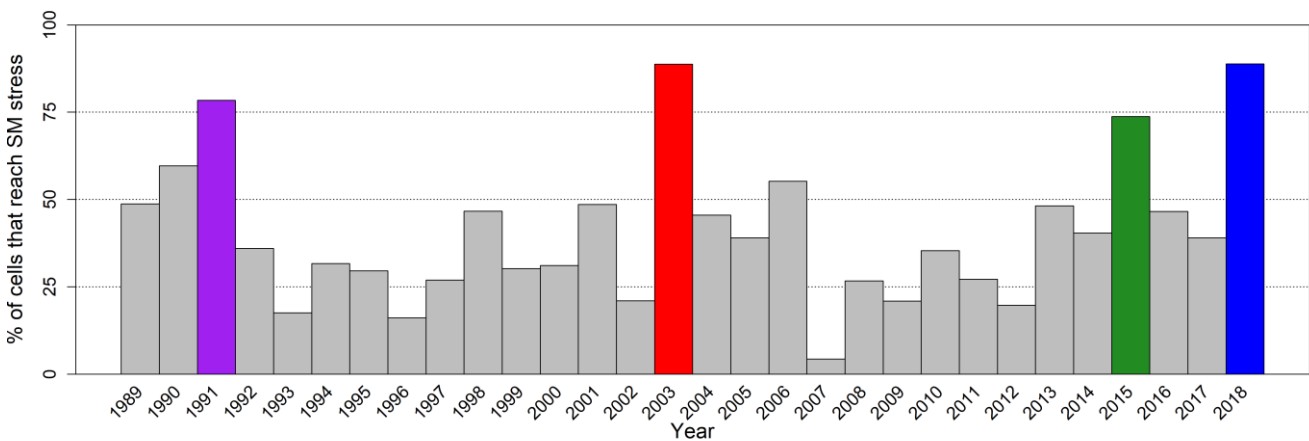

**Figure 6. Percentage of cells that reached soil moisture drought stress for at least one day ($S_{occ}$ = 1) in different calendar years. Most prominent years (1991, 2003, 2015 & 2018) are highlighted in colour.**

Figure 7 shows the relationship between the probability of reaching SM stress ($S_{occ}$) and different controls (AWC, $P_{annual}$, $T_{annual}$). In general, probability functions derived with the AWC show a steeper and annually consistent increase than probability functions derived with $P_{annual}$ and $T_{annual}$. The latter suggests a stronger influence of root zone soil characteristics, over the influence of the climatological setting, on whether or not SM stress developed. SM stress was further found to be more likely to develop in soils that have a lower AWC (Fig. 7a), as the probability of $S_{occ}$ increases with decreasing AWC.

The direction of increasing probability was consistent for every year, i.e., grid cells with a lower AWC always had a higher probability of reaching SM stress than grid cells with a higher AWC. However, during the most prominent drought years, the probability functions are shifted to the right, revealing a higher probability of reaching SM stress for grid cells with a higher AWC during these dry years. SM stress was further found to be more likely to develop in drier regions with a lower $P_{annual}$ (Fig. 7b). The probability of SM stress as a function of $T_{annual}$ shows more variation in the direction of increasing probability

(Fig. 7c). In some years, including the prominent drought years, SM stress was more likely to develop in the warmer regions, whereas in some other years, no strong relationship with temperature was observed.

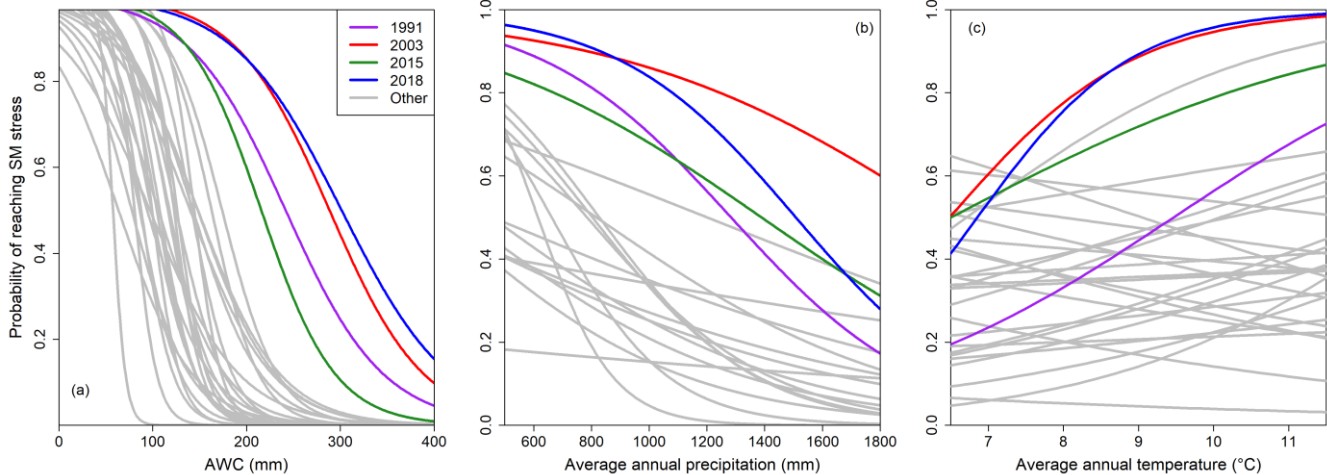

**Figure 7. Probability of reaching SM stress at least once in a year ($S_{occ} = 1$) as a function of (a) the AWC, (b) $P_{annual}$, (c) $T_{annual}$. Each curve reflects a different year. Curves of prominent drought years are highlighted in colour.**

Figure 8 shows the variation in SM stress characteristics. In general, there was a lot of within year variability in these characteristics, whereas differences between prominent drought years were often less pronounced. $S_{start}$ varies from the end of April to the end of September (Fig. 8a). The distribution of $S_{start}$ is comparable between 2003, 2015 and 2018, whereas the distribution of $S_{start}$ of 1991 indicates a generally later onset of SM stress. $S_{devtime}$ shows a large variability; from as little as 10 days to over 4 months (Fig. 8b). Despite the large within year variability of $S_{devtime}$, there were no evident differences in the development time distribution among the prominent years. $S_{total}$ shows both a large within year variability as well as distinct differences among the prominent drought years (Fig. 8c). The distribution of $S_{total}$ reveals that 2003 and especially 2018 were characterized by the longest total time in SM stress (median $S_{total,2018} = 91$ days, 95th quantile $S_{total,2018} = 151$ days). A similar within year variability and between year differences was found for $S_{maxdur}$ (Fig. 8d). Especially 2018 was characterized by persistent SM stress events (median $S_{maxdur,2018}$ of 79 days, 95th percentile of 147 days).

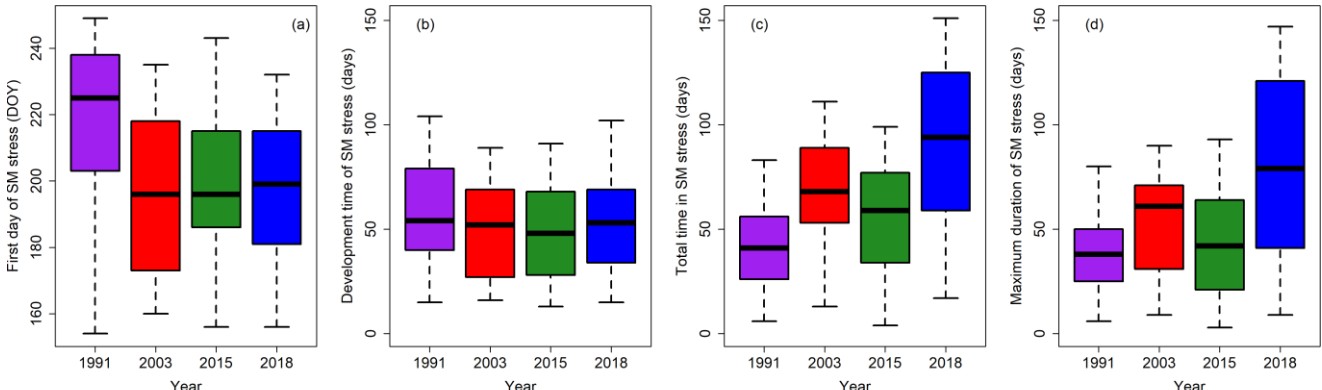

**Figure 8. Variability of different SM stress characteristics shown for the prominent drought years. Shown are (a) first day ($S_{start}$), (b) development time ($S_{devtime}$) (c) total number of days ($S_{total}$), and (d) maximum duration ($S_{maxdur}$) of SM stress. Box: percentiles 25, 50 and 75. End of whiskers: percentiles 5 and 95.**

Table 1 reveals Spearman's rank correlation coefficient between various SM stress characteristics and the AWC of the root zone as well as the climatological setting ($P_{annual}$, $T_{annual}$) during prominent drought years. Both $S_{start}$ and $S_{devtime}$ were most strongly correlated with the AWC, whereas the correlation with $P_{annual}$ or $T_{annual}$ was weaker or absent. These correlations imply that the start of soil moisture stress tends to be later and the development time tends to be longer for soils with a higher AWC. The correlations between the persistence of SM stress ($S_{total}$ and $S_{maxdur}$) and the considered soil and climate controls suggest

that the time in soil moisture drought stress tends to be longer for soils with a lower AWC that are located in drier and warmer domains of the study region. However, the correlations were weak or non-existent, and the sign of the correlation coefficient was not always consistent.

**Table 1. Spearman's rank correlation coefficient between different SM stress characteristics; first day ($S_{start}$), development time ($S_{devtime}$), total time ($S_{total}$) and maximum duration ($S_{maxdur}$), and different soil and climate controls; available water-holding capacity**
**of the root zone (AWC), annual average precipitation ($P_{annual}$) and annual average temperature ($T_{annual}$), during four prominent drought years.**

| | Year | AWC | $P_{annual}$ | $T_{annual}$ |
|---|---|---|---|---|
| $S_{start}$ | 1991 | 0.72 | 0.15 | 0.02 |
| | 2003 | 0.71 | 0.08 | -0.02 |
| | 2015 | 0.79 | 0.05 | 0.14 |
| | 2018 | 0.74 | 0.09 | -0.04 |
| $S_{devtime}$ | 1991 | 0.85 | -0.34 | 0.48 |
| | 2003 | 0.77 | -0.14 | 0.15 |
| | 2015 | 0.84 | -0.37 | 0.53 |
| | 2018 | 0.77 | -0.21 | 0.24 |
| $S_{total}$ | 1991 | -0.47 | -0.35 | 0.14 |
| | 2003 | -0.37 | -0.37 | 0.31 |
| | 2015 | -0.32 | -0.22 | 0.12 |
| | 2018 | 0.09 | -0.47 | 0.6 |

| | | | | |
|---|---|---|---|---|
| $S_{\text{maxdur}}$ | 1991 | -0.38 | -0.39 | 0.19 |
| | 2003 | 0.00 | -0.46 | 0.44 |
| | 2015 | -0.11 | -0.21 | 0.24 |
| | 2018 | 0.23 | -0.45 | 0.61 |

Figure 9 shows the meteorological anomalies during the development and annual maximum duration of SM stress (all events of all years combined; but separated based on the length of the development time and duration, i.e., shorter or longer than 30 days). During the development of SM stress, $P_{\text{perc,dev}}$ was almost always anomalously low, whereas $T_{\text{perc,dev}}$ and especially $E_{\text{perc,dev}}$ were often anomalously high, especially for the more quickly developing events (Fig. 9a). The distribution of $E_{\text{perc,dev}}$ and especially $T_{\text{perc,dev}}$ shows a larger spread than the distribution of $P_{\text{perc,dev}}$. The latter implies that especially $P$ needed to be anomalously low for SM stress to develop, whereas $E$ and $T$ could be more variable during the development. During the annual maximum duration SM stress event, $P_{\text{perc,maxdur}}$ was again generally anomalously low (Fig. 9b). However, $P_{\text{perc,maxdur}}$ shows a larger variation and spread and was generally higher than $P_{\text{perc,dev}}$, particularly for the shorter events. $T_{\text{perc,maxdur}}$ and $E_{\text{perc,maxdur}}$ show contrasting anomalies, where $T$ was often above normal and $E$ often below normal during the annual maximum duration SM stress event, especially for the events with a longer duration.

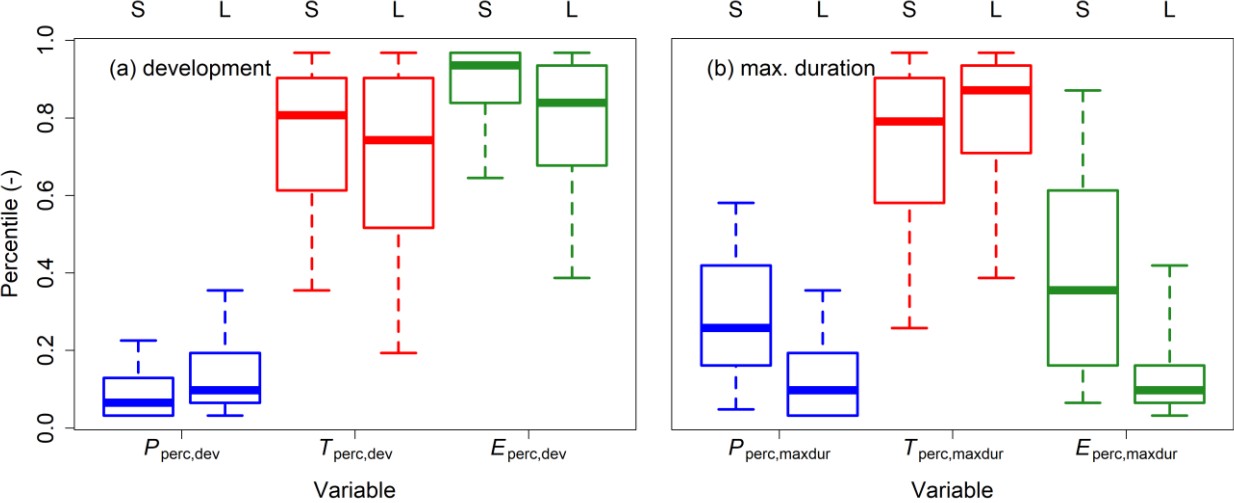

**Figure 9. Meteorological anomalies (percentiles) of precipitation ($P_{\text{perc}}$), temperature ($T_{\text{perc}}$) and actual evapotranspiration ($E_{\text{perc}}$) during (a) the development (dev) and (b) the annual maximum duration (maxdur) of SM stress. Results are split in SM stress episodes with a relatively short (< 30 days; ) and relatively long (≥30 days) development time (ratio S/L is 40/60 %) and maximum duration (ratio S/L is 67/ 33 %). Box: percentiles 25, 50 and 75. End of whiskers: percentiles 5 and 95.**

### 3.3 Sensitivity to the parametrization of $S_{\text{rootzone}}$ and used identification method

The sensitivity analyses reveal that the parameterization of $S_{\text{rootzone}}$ affected the total amount of agricultural grid cells that reach SM stress (Fig. S6). However, this parameterization had little effect on the relative ordering among drought years, i.e.,

independent of the chosen $S_{rootzone}$ parameterization, most severe drought years in terms of the amount of grid cells that reach SM stress were 2018 and 2003, followed by 2015 and 1991. Further, differences in the amount of grid cells that reach SM stress were small between results derived from simulations with a soil based $S_{rootzone}$ and a soil based $S_{rootzone}$ constrained at one-meter depth. Larger differences were found among results derived from simulations with a $S_{rootzone}$ that had a fixed AWC. More distinct were differences between SM stress and SM drought. Most grid cells reached an anomalously low state at least once in a calendar year, independent of the parameterization of the root zone, whereas SM stress shows more variation between individual years.

The probability of reaching SM stress was affected by the AWC of the root zone for results derived from soil based $S_{rootzone}$ parameterizations (Fig. S7). In case the AWC was fixed, its control was obviously removed, and the climatological setting ($P_{annual}$ & $T_{annual}$) had a larger influence. Especially results derived from a fixed AWC of 200 mm show a clear distinction where SM stress had a higher probability to develop in relatively dry and warm regions, with a shift in probability functions towards wetter and colder regions during prominent drought years. For SM drought, there was little to no relationship between the probability of reaching SM drought for at least one day in a certain year and the considered controls, given that most grid cells reach SM drought for at least one day in most years.

The parameterization of $S_{rootzone}$ also affected other SM stress characteristics ($S_{start}$, $S_{devtime}$, $S_{total}$ and $S_{maxdur}$) in their overall magnitude (Fig. S8). However, the relative ordering in the severity of prominent drought years according to those characteristics was often preserved. The distributions of $S_{start}$ were comparable between soil based $S_{rootzone}$ parameterizations, whereas $S_{start}$ is generally earlier for root zones with a fixed AWC of 100 mm and later for root zones with a fixed AWC of 200 mm (as expected). More pronounced was the difference between $S_{start}$ of SM stress and SM drought, as daily SM anomalies reached a below normal state for the first time much earlier in the year. $S_{devtime}$ also varied depending on the $S_{rootzone}$ parameterization. As expected, SM stress developed faster for root zones with the AWC fixed at 100 mm, slower for root zones with the AWC fixed at 200 mm, and somewhere in between these ranges for soil-based parameterizations of the root zone (with again little differences between the two soil based parameterizations). The ordering of the boxplots among prominent drought years were comparable among $S_{rootzone}$ parameterizations, despite for 2003 developing relatively fast with the AWC fixed at 100 mm, and 2015 developing relatively slow with the AWC fixed to 200 mm. The distributions of the $S_{total}$ and $S_{maxdur}$ in different drought years were comparable among $S_{rootzone}$ parameterizations. More notable is the difference with SM drought, i.e., SM was much longer (continuously) in an anomalously low state as compared to the time in SM stress. The ordering of most severe drought years according to duration of the drought remained the same and is comparable to the ordering of SM stress, with one notable difference for the drought of 2003 derived from a fixed $S_{rootzone}$ parametrization with an AWC of 200 mm, which lasted relatively long compared to other drought years.

$S_{start}$ of SM stress derived from the simulations with a soil based root zone parameterization was most strongly related to the AWC and less to the climatological setting (Fig. S9). When the AWC of $S_{rootzone}$ was fixed, $S_{start}$ positively correlated with $P_{annual}$ and negatively with $T_{annual}$ i.e., SM stress started later in wetter and colder regions. This is different for SM drought (anomaly), of which the first day is positively correlated with $P_{annual}$ and negatively correlated with $T_{annual}$. $S_{devtime}$ of SM stress

is most strongly correlated to the AWC, and less to the climatological setting for soil-based root zone parameterizations. For root zone parameterizations with a fixed AWC, no correlations with $P_{annual}$ and a negative correlation with $T_{annual}$ (some years) were found. $S_{total}$ and $S_{maxdur}$ of SM stress were only weakly correlated to the AWC of the root zone, whereas the total and maximum duration of SM drought showed a strong positive correlation with the AWC. In other words, SM droughts lasted much longer in thicker root zones with a higher AWC whereas these root zones were not necessarily in a longer state of SM stress. $S_{maxdur}$ and $S_{total}$ of SM stress were further correlated to $P_{annual}$ and $T_{annual}$, especially for (shallow) root zones with a fixed AWC, whereas $S_{maxdur}$ and $S_{total}$ of SM drought generally showed lower correlations with $P_{annual}$ and $T_{annual.}$

## 4 Discussion

Our first objective was to characterize the occurrence, development time and persistence of simulated past soil moisture (SM) stress events. Results revealed a large temporal variability in the amount of grid cells that reach SM stress in a certain year (Fig. 6). The most severe SM stress years were 2003 and 2018, during which up to 89 percent of the agricultural grid cells reached SM stress. These percentages of grid cells were found to be (slightly) different depending on the parameterization of $S_{rootzone}$ (Fig. S6), implying differences between e.g. shallow rooting crops with limited access to water and deeper rooting crops with a larger water availability. Nevertheless, the ordering of most severe drought years was not affected by the parameterization of the root zone, i.e., 2003 and 2018 were always characterized as most severe in terms of the amount of grid cells that reached SM stress. Previous studies already showed that 2003 was an extreme drought year within and around the study region (e.g., Schär & Jendritzky, 2004; Ionita et al., 2016). Results of this study imply that the recent 2018 event was comparable to 2003 in terms of the amount of grid cells that reach SM stress. However, even during these most severe drought years, SM stress did not develop for some of the agricultural grid cells (unless a root zone with a fixed AWC of 100 mm was used), either because of 1) local variations in meteorological conditions (e.g. local rains storms) and b) root zone soils having a large enough storage capacity that acted as a buffer during dry conditions. This illustrates that even during the most extreme drought years, regional differences can occur. The factors that control these differences, i.e., the occurrence of local rainstorms and differences in soil characteristics can be spatially heterogeneous. The latter implies that regional agricultural drought assessments and monitoring should occur at a relatively high spatial resolution to be able to capture these differences.

A large variability in the development time of simulated SM stress was found (Fig. 8b). SM stress could develop in less than 10 days, e.g., in shallow root zones with a low available water-holding capacity (AWC). This is faster than the minimum development time of 30 days used to identify rapid-onset (flash) droughts in, e.g. Christian et al. (2019). On the other hand, it could also take a lot longer (over 4 months) for SM stress to develop. This slower development matches better with the traditional description of drought, being a slowly developing (creeping) phenomena (Wilhite & Glantz, 1985). The above-stated (ranges in) development time were reduced when the starting point of SM stress development was set to a level lower than field capacity (Fig. S10), implying that it is important to keep track of partially depleted soil moisture stores that can be a precursor of more rapid development. The sensitivity analyses revealed that fixing $S_{rootzone}$ reduces the variability in

development time; however, also showing the distinct differences between root zones with a relatively low and high storage capacity, indicative for differences between shallow and deep rooting crop species. Further, the relative ordering of drought years in terms of their $S_{devtime}$ was often the same, besides few exceptions, which relates to specific differences in the configuration of meteorological dry spells and whether they caused SM stress under different $S_{rootzone}$ parameterizations. Overall, the large differences in development time suggest that different types of forecasting systems could be suitable to predict the development of SM stress; medium range weather forecasts for quickly developing events and more long-term meteorological forecasts for slower developing episodes.

The persistence of SM stress (total days and maximum duration) varied strongly between years and grid cells (Fig. 8c,d). Results of this study showed that the total days and maximum duration of SM stress was generally highest in 2018, making this event more severe than earlier (recent) benchmark events, such as 2003. The long nature of the drought of 2018 was also found in a recent study for Switzerland, the country directly south of our study region, in Brunner et al., (2019). The ordering of most extreme drought years according to duration was often found to be independent of the parameterization of $S_{rootzone}$ or whether SM stress or drought was analyzed (Fig. S9). On the other hand, distinct differences in duration were found, especially between SM stress and drought, i.e., SM was generally much longer in an (continuous) anomalously low state compared to the time it was in a state of SM stress. This can partially be explained by the fact that SM can be anomalously low without being severely depleted, especially towards the end of the year after a severe drought year when SM stores are not completely filled to field capacity again (as would normally be the case). We also found that the annual maximum duration and total time of SM stress never exceeded 6 months, and most of the root zones reached field capacity again each year before the start of the new growing season. Thus, SM stress was never a multi-year phenomenon for the considered agricultural grid cells. SM drought on the other hand, can last longer and could more easily persist into the next year.

Our second objective was to investigate the dominant controls on the probability, development time and persistence of SM stress. Both probability and development time were most strongly related to the AWC of the root zone and less to the climatological setting (Fig. 7, Table 1). SM stress was generally more likely to develop, and it evolved faster and earlier in the year in shallow root zones with a lower AWC. These findings are in line with results for the 2012 flash drought in the US presented by Otkin et al. (2016), where anomalous soil moisture conditions generally first appeared in the topsoil layer (lower AWC) and only later in the entire soil layer (higher AWC). Results also confirm that AWC of the root zone is an important factor to determine the vulnerability to agricultural drought, as was also stated in, e.g., Wilhelmi & Wilhite (2004). Here, it is important to state that AWC is not only a soil parameter but also encompasses differences between e.g. a shallow or deep rooting crop, as was exemplified by the differences between the two root zone parameterizations with a fixed AWC found in the sensitivity analyses (Fig. S6, S8). Finally, these results imply that agricultural drought assessments purely based on meteorological proxy indicators should be interpreted with care, as most meteorological proxy indicators do not consider differences in root zone soil characteristics.

The persistence of SM stress was only weakly correlated with the AWC of the root zone and more strongly with the climatological setting (Table 1), especially when considering a parameterization of $S_{rootzone}$ with a fixed AWC (Fig. S9). The

reason for the overall weaker correlations with the AWC might be related to the different mechanisms that govern the persistence of SM stress in different types of root zones. In root zones with a low AWC, SM stress can develop rather quickly. However, the total deficit that can build up is limited and only a small rainfall event is enough to alleviate SM stress conditions.

In root zones with a high AWC, larger SM deficits can potentially develop. However, this development takes longer, and the SM stress threshold is only exceeded towards the end of the growing season, after which further development is limited because of lacking evapotranspiration. The most persistent SM stress events might therefore occur for root zones with an intermediate AWC. In these root zones, SM stress can develop reasonably fast but can also build up a large enough deficit that can endure some smaller rainfall events. This is different for the duration of SM drought (anomaly) which is positively correlated with the

AWC of the root zone, i.e., SM droughts tend to last longer for root zones with a higher storage (Fig. S9). One reason for this is that SM (anomaly) time series derived from root zones with a larger AWC often exhibit a much more gradual behavior, whereas SM (anomaly) time series derived from root zones with a smaller AWC are often flashier. Another reason for this is that it can take much longer for root zones with a larger AWC to reach a level of field capacity towards the end of the year (normal conditions) after a prolonged meteorological dry spell.

The third objective of this study was to portray the meteorological anomalies during (the development of) simulated SM stress. During the development, especially precipitation needed to be anomalously low, particularly during the more rapid developing events (Fig. 9a), suggesting that lacking precipitation was the most important prerequisite for SM stress to develop. However, also air temperature and evapotranspiration were often above normally high during the development of SM stress, implying an enhancing (compound) effect of these variables (see also Manning et al., 2018), especially during rapid onset events. During

the annual maximum duration SM stress events, precipitation was often below normal as well, especially for the longer events (Fig. 9b). However, precipitation anomalies during the maximum duration events were not as extreme as during the development, possibly because SM only needed to remain in a steady state condition of SM stress rather than having to decline from field capacity to a level of SM stress. Temperature and simulated evapotranspiration show contrasting anomalies during the annual maximum duration SM stress events, with temperature generally being above- and simulated evapotranspiration

generally being below- normal, particularly during the longer events. The reason for these contrasting anomalies might be related to a different energy partitioning of heat fluxes during SM stress (described in e.g., Seneviratne et al., 2010). During SM stress, simulated evapotranspiration was anomalously low because of the water stress for vegetation that causes plants to limit their evapotranspiration assumed in the model. The incoming solar radiation that is normally consumed by evapotranspiration (latent heat flux) is now used to warm up the soil and lower atmosphere (sensible heat flux), possibly

explaining the above normal temperatures during SM stress (Miralles et al., 2014). This energy partitioning during SM stress and resulting contrasting temperature and evapotranspiration anomalies highlight that agricultural drought assessments derived from meteorological proxy indicators based on potential evapotranspiration should be interpreted with care.

Our regional assessment of SM stress is subject to inaccuracies, challenges, and assumptions; something common for these kinds of analyses. One source of inaccuracies relates to the modeling of SM. Previous studies showed that the physical based

TRAIN model was able to provide a good temporal representation of soil moisture over agricultural fields (e.g., Stork &

Menzel, 2016). However, it is important to bear in mind that the studied results are regional model simulations for specific soil and land use parameterizations that can differentiate from the heterogeneous real world. An evaluation of the simulated hydrological fluxes with observed streamflow suggests that TRAIN provides a reasonable estimation of the water balance and its variability (Fig. 5, S2-S5). However, there are other models, model structures and model parameterizations to simulate soil
moisture, implying a dependency between the used model (parameterization) and the results (shown in e.g., Samaniego et al., 2018; Zink et al., 2017). The latter studies use ensembles of resp. different models or different model parameterizations to consider model or parameter related uncertainties; something outside the scope of the current study.

Another source of inaccuracies stems from the data used to set-up and force the model. One challenge was the interpolation of several different meteorological variables over a rather complex terrain, which is prone to biases, especially for variables such
as wind speed. Another challenge was the spatially accurate representation of the root zone soil, both in terms of the interpolation of heterogeneous soil and land use characteristics as well as in the parameterization of the rooting depth. The interpolation of soil and land use characteristics was based on the majority class within a 1-km grid cell. However, each grid cell can still exhibit a large variability in soil and land use characteristics, implying that the simulated SM dynamics might not be representative for the entire grid cell. The parameterization of the rooting depth of each grid cell was further based on soil
characteristics, which is a more often used procedure to parameterize regional models. However, roots do not necessarily utilize the water in the entire soil column, and rooting depth is depending on other factors such as the type of crop. For example, a soil might have a maximum rooting depth of a meter; however, if a shallow rooting crop species is grown on this soil, roots may not have access to all water. A sensitivity analyses revealed that derived results change depending on the used parameterization. Differences in (controls on) simulated SM stress characteristics are small between a soil-based root zone
and a soil-based root zone constrained to one-meter depth, implying that the latter depth-constrain does not have a great impact on simulated SM stress characteristics. Larger were differences when the volume of the AWC of the root zone was constrained to a fixed value, i.e., mimicking shallow and deeper rooting crop species with resp. lower and higher water availabilities. A not considered option was a climate-based parameterization of the root zone, which works with the hypothesis that the (catchment-average) size of the AWC of $S_{rootzone}$ (dynamically) develops to deal with meteorological droughts of certain return
periods (e.g. 10 years). Various studies show improved model performances for a selection of catchments when defining the root zones in such a way as opposed to a soil based definition (e.g. de Boer-Euser, et al., 2016). The reason why we did not apply this parameterization in our study is that 1) we focus on annual agricultural crops, that are harvested every year and thereby do not have the opportunity to gradually adapt their root zones over time, 2) such analyses requires a study with a different scope. In the end, an accurate spatiotemporal representation of the root zone, considering the influence of soil-,
climate- and crop sf. With the sensitivity analyses, we cover four possible scenarios, but different assumptions might apply depending on the scope of the study.

The soil-based parametrization of the root zone, the variability of soil and land use characteristics within a single grid cell as well as possible biases in interpolated meteorological variables means that results might not always be accurate for a specific grid cell or for a single agricultural field located within this grid cell. However, by analyzing a large sample of grid cells and

by including a sensitivity analyses to the parameterization of the root zone, we cover a large number of combinations of root zone characteristics and climatological settings that occur within the study region (Fig. 1). Lessons learned from these large samples, e.g., about the relationship between SM stress characteristics and soil or climate properties (e.g. Fig. 7, Table 1), provide insights that might be relevant for smaller (local) scales within the study region, however, only when the modeling assumptions, e.g., behind the parameterizations of $S_{rootzone}$, apply. Here, the most suitable assumption can vary depending on the studied crop, e.g., whether a crop is studied that makes full use of all plant-available water in the root zone soil or a shallow rooting crop that only uses of part of it.

An assumption that was made in this study relates to the definition of SM stress. We characterized periods of SM stress (absolute) rather than SM drought (anomaly). We used one fixed threshold of 30% of the AWC to define SM stress. This threshold is in line with the indicative threshold for potential SM stress used by the German Weather Service (DWD, 2018). However, it should be noted that this threshold, as well as the relationship between the degree of SM stress and the amount of available water left in the root zone, varies depending on, e.g., crop species, climatological conditions and soil type (Allen et al., 1998). Notwithstanding these assumptions, we believe that from an agricultural drought impact perspective, the used definition of SM stress is more closely related to actual water stress experienced by plants than an anomaly-based definition. Especially so, because SM anomalies can be significantly different from SM stress and below normal anomalies often correspond to situations of sufficient soil moisture (Fig. 10). SM stress often still relates to an anomalously low state that develops and persist during periods with below normal precipitation (Fig. 9). However, SM stress also incorporates temporal variability with an increased occurrence during the growing season and a limited occurrence during the non-growing season, whereas SM drought occurs equally distributed over the year (Fig. 10). Further, rareness of SM stress is affected by the plant available water-holding capacity of the root zone soil and the climatological setting as revealed by e.g. the ranges in Fig. 10 or the difference between Fig. 10c and d, whereas this is not the case for SM drought. On the other hand, it should be noted that derived SM stress characteristics are more sensitive to modeling assumptions and uncertainties. SM stress characteristics derived from simulations using different parametrizations of the root zone reveal more variation (Fig. 10) but therefore also a higher degree of disagreement on whether SM stress was reached (Fig. S11a). Soil moisture anomalies show a higher degree of agreement, i.e., results are more robust and much less sensitive to the (uncertainties in) parameterization of the root zone (Fig. S11b). Overall, the in this study used definition of SM stress might be applicable in other regions or for other research purposes, e.g., that aim to investigate changes in agricultural drought vulnerability under climate change.

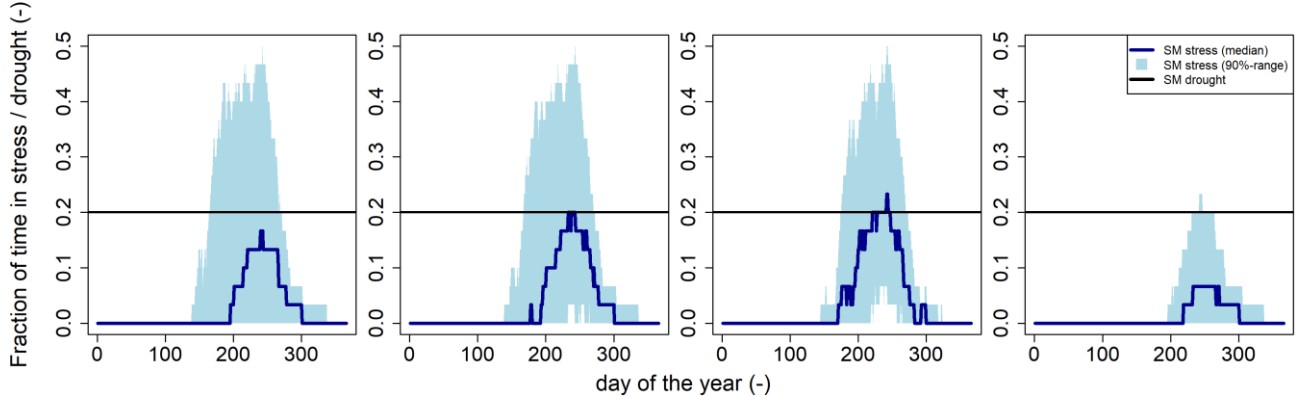

**Figure 10. Temporal variation in SM stress and drought occurrence frequency derived for each day of the year from results of different parameterizations of the root zone (a) soil based, (b) soil based constrained at 1 meter depth, (c) $S_{rootzone}$ with a fixed AWC of 100 mm and (d) $S_{rootzone}$ with a fixed AWC of 200 mm.**

## 5 Conclusion

Meteorological droughts cause soil moisture levels to decline. Diminished root zone soil moisture can largely affect agricultural productivity, as crops might experience soil moisture stress. In this study, we investigated the characteristics of simulated past soil moisture stress events across the agricultural regions of Southwestern Germany as well as their relationship with soil and climate variables. The total agricultural area that reached soil moisture stress conditions was found to vary strongly among the years and was highest in 2003 and 2018. In terms of the development time, 2003 was not much different from 2018. In both years, development time varied from as little as 10 days to over four months. What made 2018 distinctively different from 2003 was the generally longer total time and maximum duration of simulated soil moisture stress, highlighting the extraordinary severity of the most recent event studied.

Both the occurrence and development time of soil moisture stress were found to be strongly related to the available water-holding capacity of the root zone and not so much to the climatological setting. That is, when we assume roots can make use of all available water in the root zone column either or not constrained at a depth of 1 meter. When we assume root zones of fixed sizes, the influence of the climatological setting increases, yet the difference between a shallower rooting crop (lower AWC) and a deeper rooting crop (higher AWC) remain. Thus, the above findings stress the importance of considering differences in root zone storage characteristics for agricultural drought assessments and monitoring and early warning, independent on whether these differences in storage are related to the difference in soil or crop species. Nonetheless, a major challenge remains the accurate spatial-temporal characterization of the root zone soil that considers (the interactions between) soil, climatological, meteorological and crop specific factors.

Results of this study further imply that below normal precipitation was the most important reason for soil moisture stress to develop. However, the often above normal anomalies of temperature and especially simulated evapotranspiration during

development, suggest an augmenting effect of these variables. During soil moisture stress, temperature anomalies were found to be often above normal, which contradicted with the often below normal simulated evapotranspiration anomalies. These contrasting anomalies of temperature and evapotranspiration imply that agricultural drought assessments derived from meteorological proxies based on potential evapotranspiration should be interpreted with care. The same is the case for agricultural assessments based on soil moisture anomalies, as below normal anomalies were found to not necessarily correspond to a situation of soil moisture stress, especially for periods outside the growing season. In addition, the sensitivity analyses revealed that SM drought characteristics, and controls on these characteristics, can differ significantly from (controls on) SM stress characteristics. Overall, the in this study presented approach of directly characterizing simulated soil moisture stress events for agricultural drought assessments might in some cases be a suitable alternative to approaches based on meteorological proxies or soil moisture anomalies.

**Code and data availability.** Gridded model simulations of soil moisture used in this study as well as animations of the latter during major drought events are available from the Heidata repository of the Heidelberg University. The following DOI is reserved and will become active upon acceptance https://doi.org/10.11588/data/PRXZAS. For reviewing purposes, the data is accessible via the following link https://heidata.uni-heidelberg.de/privateurl.xhtml?token=fb658f7f-0ec8-49db-84d0-a8e726936743). Input data for the model can be derived from publicly available sources (Section 2.2). The used Models and R-code can be obtained from the authors upon request.

**Author contributions.** ET and LM designed the study. ET prepared the data, carried out the analyses, wrote the manuscript and prepared the Figures and Tables. LM provided input on the analyses and edited the paper.

**Competing interests.** The authors declare that they do not have competing interests.

**Acknowledgements.** This work contributes to the DRIeR project funded under the framework of the "Research Network Water" by the Ministry of Science, Research, and the Arts of the German Federal State of Baden-Württemberg. We thankfully acknowledge Verena Maurer for her help with interpolating the soil and land cover grids, Anna Buch for testing and preparing the SARAH global radiation data as TRAIN input, Michael Stoelzle for providing a list of catchments with near-natural flow located in Baden-Württemberg and Nicole Gerlach for her help with the INTERMET Software. We further acknowledge all agencies that provided the data used for the simulations, specifically the Federal Agency for Cartography and Geodesy (BKG), the German Environment Agency (UBA) as well as the Environment Agency of Baden-Württemberg (LUBW), the Federal State Office for Geology Resources and Mining (LGRB), the German Weather Service (DWD) and the Satellite Application Facility on Climate Monitoring (CM SAF). Financial support of the DFG for data storage is acknowledged. All analyses were carried out with the open-source software R (https://www.r-project.org/), partially using the packages "raster ,"rgdal" and "rdwd".

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
