# Peer review of "The development and persistence of soil moisture stress during drought across Southwestern Germany"

_Hydrology and Earth System Sciences, 2020_

## Short Comment (SC1) · 12 Jul 2020

It is interesting to read the entitled paper "Controls on the development and persistence of soil moisture drought across Southwestern Germany". The manuscript is well-written and well-organized. I have a few suggestions regarding the method.

It seems the authors used the linear correlation and regression models to identify the individual contribution from different controls on SM drought features. However, these two approaches can not differ the co-influencing between the controls (i.e., the soil properties and climate settings). Probably, the partial least squares regression (PLSR) and the partial correlation analysis could be more efficient to identify the individual

impacts for each control by excluding other controls' effect. Kindly see the R function, e.g., plsr and pcor.test in R program. The results based on the PLSR and partial correlation could be different from the current results. The Authors could make some tests based on their sample data. It is only a suggestion.

---

## Short Comment (SC2) · 22 Jul 2020

(1) The paper by Tijdeman and Menzel investigated the spatial and temporal variability of soil moisture drought in southwestern Germany using a distributed hydrological model. They analyzed the environmental controls on drought development and duration. Some interesting results are presented in the paper. For example, they find that drought stress tends to occur in warmer and drier locations. This raises an important question, i.e. how did and will climate change affect drought occurrence and severity? Some researchers have conducted some attempts to answer this question at a large scale, e.g. Samaniego et al. (2018). I would suggest the authors do some work on

this question in the future. For example, the authors may simulate the drought stress at different warming and precipitation change scenarios, and see what land covers will be affected most. I know that this analysis is already beyond the scope of this paper. The authors do not need to add this analysis in this paper. I have some other comments below, which may be helpful for the authors to improve this manuscript.

Samaniego, L., Thober, S., Kumar, R. et al. Anthropogenic warming exacerbates European soil moisture droughts. Nature Clim Change 8, 421–426 (2018). https://doi.org/10.1038/s41558-018-0138-5

(2) L102. Figure 1a. I would suggest clipping the elevation map and only reserve the Baden-Württemberg. Then you may add a small panel at the corner which indicates the location of Baden-Württemberg in Germany.

(3) L169. In this paper, the SM drought threshold is set to 30% of AWC. Then a binary time series of SM drought stress occurrence becomes the basic data of this study. I am thinking it might be helpful to further classify the SM drought to different levels, for example, moderate, severe and extreme SM droughts. In this case, Figure 4 may demonstrates the temporal variations of cell counts for different drought severity. It may provide some information like whether climate warming has increased the drought severity in this region.

(4) L170-175. SM drought stress occurrence (Socc) was computed in the basis of calendar year in this paper. Normally, most of the soil moisture droughts in Germany happen between spring and autumn. However, was there some winter droughts over 1989-2018, which began at the end of a year and ended in the following year? If yes, these special circumstances may overestimate the drought occurrences in the successive two years. In addition, how did you calculate the development time and duration for these special droughts? I assume these events are very rare in this region.

(5) L225-227. This sentence is confusing. What does "the latter" refers to? What I see is that drought tends to develop at warmer locations for all prominent drought years,

but not for all the other years. Please make it more clear.

(6) L255-258. It would be helpful to add the significance test of the rank correlations in Table 1.

---

## Referee Comment (RC1) · Anonymous Referee #1 · 29 Jul 2020

**Review of Tijdeman and Menzel: Controls on the development and persistence of soil moisture drought across Southwestern Germany**

**General comments**

This manuscript investigates the role of soil characteristics in the root zone and climate properties in determining the probability of occurence and characteristics of agricultural drought. The manuscript is well written albeit sometimes rather lengthy and repetitive. Analyses are systematic as is the presentation of the results. The main conclusion to my understanding is that root zone storage characteristics are important for agricultural drought assessment and people should not only look at meteorological metrics. Although I fully agree with that statement irrespective of the results in this study, I do not think that this statement is justified based on this study's findings. My main issues are:

- The data used for the available water-holding capacity (AWC, i.e., the amount of plant available water in the root zone at field capacity), might not be representative for the actual amount of water available to vegetation at all and could be significantly biased as climate and land cover types are in reality the main controls on root zone storage capacity and not the soil type. This would be fine, however, if we would accept that AWC is simply a soil characteristic, but then the definition of soil moisture stress occuring at 30% AWC might be biased instead.
- Conclusions are drawn on AWC being a control of reaching 30% AWC. This is clearly circular reasoning and those findings can hardly be considered surprising.
- This study evaluates the soil moisture within a hydrological model (TRAIN), however, there is no information shown on the setup of the model and whether this model performs well at all based on streamflow or other measurements. This might be shown in the papers that are referred to, but I would find it useful here as well. Neither is it evaluated how crucial information/parameterization affects the results. Does the in- or exclusion of the AWC data vs. a fixed value improve model performance? Is the vegetation water stress formulation in TRAIN really the best and would other parameters lead to worse or better streamflow predictions?

Conclusion: although the research is systematic and well presented, and I do not have a lot of comments, I personally do not see how the authors would be able to address these comments without adding new analyses. Doing so, would make the revision deviate considerably from the original submission and, therefore, I would recommend a rejection with the encouragement to resubmit, rather than major revisions. I believe that although this is a harsh recommendation it would also be in the interest of the authors themselves to have a revised manuscript evaluated starting from the public discussion phase.

**Specific comments**

**L38: "Droughts are often defined as a below normal water availability"**
I would have expected some critical reflections on this directly in or directly after this paragraph and not by the end of the introduction.

**L75: "which is indicative for low soil moisture levels causing drought stress for plant"**
Given the fact that at this point in the introduction drought has only be described to be defined as an anomaly and not as an absolute measure, low soil moisture levels can occur without having a drought, so the plants in this example just experience water stress and not drought stress.

**L109: "Vectorized soil property data (field capacity and wilting point of the root zone soil) were derived from the BK-50 (scale of 1:50,000) dataset provided by the Federal State Office for Geology Resources and Mining (LGRB, 2019)."**
Is this the available water-holding capacity in the rootzone? Does it include thickness as well as soil type? This is not clear. More importantly: how do you know that plants' roots really access all this water? There have been many studies showing that the root zone storage capacity is not a characteristic of the soil, but mainly that of the climate and the plant (e.g., de Boer-Euser et al., 2016; Fan et al., 2017; Gao et al., 2014; Guswa, 2008; Kleidon, 2004; Nijzink et al., 2016; Speich et al., 2018). Therefore, it should be made clear in the manuscript that AWC is a soil property within a part of the rootzone, but not necessarily a characteristic of the rootzone itself, and may even be completely unrelated to root zone water storage capacity.

**L145-146: "Thus, the root zone soil is not subdivided into different layers but understood as one uniform soil column."**
Does it have a specific pre-defined thickness? Was it calibrated on something? This is a crucial parameter, so a more comprehensive description would be useful to the reader.

**L218-L220: "The latter suggests a stronger influence of root zone soil characteristics, over the influence of the climatological setting, on whether or not SM drought stress developed. SM drought stress was further found to be more likely to develop in soils that have a lower AWC (Fig. 5a), as the likelihood of Socc increases with decreasing AWC."**
Yes, obviously this is the case. The probability of occurrence of SM drought stress (defined as <30% of AWC!) is related to AWC. It's extremely obvious that these variables are related, so it's not surprising at all to find a strong relation, especially as this is an entirely model-determined results. This is clearly circular reasoning and can hardly be considered surprising.

**L302-L303: "SM drought stress was generally more likely to develop, and evolved faster and earlier in the year, in shallow root zones with a lower AWC."**
Yes, obviously this is the case as SM drought stress is defined as <30% of AWC! This is again clearly circular reasoning and can hardly be considered surprising.

**L305-L306: "Results also confirm that AWC of the root zone is an important factor to determine the vulnerability to agricultural drought"**
In your model that is and with a definition where agricultural drought is defined as a percentage of AWC. This conclusion is, therefore, overstated and should be withdrawn in case it cannot be backed up with any observations (crop yields, vegetation observations, etc.) or hard proof that the hydrological model is a reliable descriptor of true states and fluxes.

L352: "However, roots do not necessarily utilize the water in the entire soil column"
Exactly! Or they are able to access more water than what you think based on the soil map and model parameterization. There would likely be great differences between forests, grasses and crops and the roots would develop differently under different climates. Therefore, what you define as soil moisture drought stress could be far from reality.

L357: "However, by analyzing a large sample of grid cells, we cover most combinations of root zone characteristics and climatological settings that occur within the study region"
Even if we accept that the rootzone characteristics and climate to be wrongly represented in individual grid cells, you have no basis to claim that the probability distribution function of root zone vs. climate is representative of reality.

**Technical corrections**
L34: "aerial overview"
What does this mean? Aerial in the literal sense or as a figure of speech? Perhaps just use overview.

L71: "it's"
its

Fig 1. The numbers on the side are probably some kind of coordinates, but not defined. Moreover, all text is really small and difficult to read.

L114: "watt/m$^2$"
Just an example, but notation should be W m$^{-2}$ (please look at HESS Mathematical requirements)

L123: "T, Uspeed, RH and RG"
Just an example, please avoid acronyms where a single symbol could easily be used and use italic notation for physical quantities (please look at HESS Mathematical requirements)

Fig. 3. Units missing in the legend.

L204-L205: "For ease of notation, we omit the grid cell and year identifiers (i and y) from the variable subscripts in the remainder of this paper."
I don't think it was necessary then to introduce i in the first place. Moreover, y is used in the remainder of the manuscript making the statement incorrect.

Fig. 5 and beyond. What is defined here as likelihood should be probability. There is no hypothesis testing or anything that would justify using the term likelihood.

L229. "at least once in a year (Socc = 1)"
The symbol of at least once is ≥ and not =

L330: "vegetative stress"
Water stress for vegetation

**References**

de Boer-Euser, T., McMillan, H. K., Hrachowitz, M., Winsemius, H. C. and Savenije, H. H. G.: Influence of soil and climate on root zone storage capacity, Water Resour. Res., 52(3), 2009–2024, doi:10.1002/2015WR018115, 2016.

Fan, Y., Miguez-Macho, G., Jobbágy, E. G., Jackson, R. B. and Otero-Casal, C.: Hydrologic regulation of plant rooting depth, Proc. Natl. Acad. Sci., 201712381, doi:10.1073/pnas.1712381114, 2017.

Gao, H., Hrachowitz, M., Schymanski, S. J., Fenicia, F., Sriwongsitanon, N. and Savenije, H. H. G.: Climate controls how ecosystems size the root zone storage capacity at catchment scale, Geophys. Res. Lett., 41(22), 7916–7923, doi:10.1002/2014GL061668, 2014.

Guswa, A. J.: The influence of climate on root depth: A carbon cost-benefit analysis, Water Resour. Res., 44(2), 1–11, doi:10.1029/2007WR006384, 2008.

Kleidon, A.: Global datasets and rooting zone depth inferred from inverse methods, J. Clim., 17(13), 2714–2722, doi:10.1175/1520-0442(2004)017<2714:GDORZD>2.0.CO;2, 2004.

Nijzink, R., Hutton, C., Pechlivanidis, I., Capell, R., Arheimer, B., Freer, J., Han, D., Wagener, T., McGuire, K., Savenije, H. and Hrachowitz, M.: The evolution of root-zone moisture capacities after deforestation: A step towards hydrological predictions under change?, Hydrol. Earth Syst. Sci., 20(12), 4775–4799, doi:10.5194/hess-20-4775-2016, 2016.

Speich, M. J. R., Lischke, H. and Zappa, M.: Testing an optimality-based model of rooting zone water storage capacity in temperate forests, Hydrol. Earth Syst. Sci., 22(7), 4097–4124, doi:10.5194/hess-22-4097-2018, 2018.

---

## Author Comment (AC1) · 24 Aug 2020

**Reply to the comments of reviewer 1**

"This manuscript investigates the role of soil characteristics in the root zone and climate properties in determining the probability of occurence and characteristics of agricultural drought. The manuscript is well written albeit sometimes rather lengthy and repetitive. Analyses are systematic as is the presentation of the results. The main conclusion to my understanding is that root zone storage characteristics are important for agricultural drought assessment and people should not only look at meteorological metrics. Although I fully agree with that statement irrespective of the results in this study, I do not think that this statement is justified based on this study's findings. My main issues are:"

We would like to thank the reviewer for the critical remarks on our study and its assumptions, which will certainly help to improve the manuscript significantly. In summary, we fully agree that an overall evaluation with streamflow observations was missing and suggest adding such analyses carried out for 50 catchments with near-natural flow across the study region as supplementary material (see reply to major remark 3). In addition, we agree that (spatial) simulation analyses are prone to uncertainties related to assumptions made, in our case regarding the parameterization of the root zone and used drought definition and threshold. Therefore, we will include additional sensitivity analyses, comparing our approach with other approaches, although we believe that there are advantages of our approach in this regional assessment (see reply to major remark 1). Further, we realize that using the term "soil moisture drought stress" is not the most suitable. This because of the above stated uncertainties and because of the fact that we do not simulate specific plant species, which have specific (growth stage related) stress levels, but rather a more general agricultural land use class. We will refer to reduced soil moisture (SM) availability instead and emphasize that this is simulated. Nonetheless, we see some advantages of characterizing periods of reduced SM availability compared to periods with below normal soil moisture anomalies, which we will clearly outline in a new manuscript (see Reply to major remark 1). We will also weaken claims about controls on reaching a state of reduced SM availability and its development time and duration, emphasize that these are model based, and discuss how controls change under different modelling assumptions (related to major remark 2). In addition, we can remove "controls" from the title, as it is one of the conclusions, but not intended as the main one. Other conclusions relate to e.g. the characteristics of events with reduced SM availability and their development, including the drought of 2018.

Below, we will reply in blue to the comments of the reviewer (numbered for referencing purposes). However, before we do so, we would like to make two clarifying remarks related to specific comment #3.

- The available water holding capacity of the potential root zone soil (AWC) was derived from a soil dataset (Section 2.2). This dataset is based on extensive field investigations on soil profiles distributed over the whole of Germany, which led to a detailed soil map, including information about soil types, grain size distribution, sequence and depth of soil horizons as well as parameters describing water-holding capacity (field capacity, wilting point, air potential). In addition, it includes information about the potential depth of the root zone constraint by e.g. the occurrence of a root restrictive layer (broadly ranging between a few decimetres up to two meter). To make sure the reader is aware of this, we will provide the more accurate description above in Section 2.2 and Section 2.3. In addition, we will make sure to emphasize already in the method section that this is the potential root zone and that agricultural crops might not

make full use of it as well as that the potential root zone of an individual grid cell might not be representative for plot scale observations.

- The focus of the current study is on agricultural grid cells, which are all (parameterized as) annual crops (see e.g. Section 2.3, Line 164-166). These annual crops are different from perennial ecosystems such as forests and grasslands, which can gradually develop and adapt over the years – possibly optimizing their root zone systems to deal with droughts of certain return periods (as presented in part of the references provided in specific comment #3).

**Major comment #1**

"The data used for the available water-holding capacity (AWC, i.e., the amount of plant available water in the root zone at field capacity), might not be representative for the actual amount of water available to vegetation at all and could be significantly biased as climate and land cover types are in reality the main controls on root zone storage capacity and not the soil type. This would be fine, however, if we would accept that AWC is simply a soil characteristic, but then the definition of soil moisture stress occuring at 30% AWC might be biased instead. "

**On the made assumptions**

We acknowledge that our soil-based definition of the potential root zone might not be representative of the actual root zone. However, regional assessments rely on certain assumptions, and we prefer a soil based root zone definition in this study, as it at least takes into account the variability in potential rooting depth and available water holding capacity (AWC). In addition, the occurrence of a rootrestricting layer in the soil has shown to influence root development (Schneider and Don 2019), and is more often used as one of the boundary conditions in crop or hydrological models (Gayler et al. 2014; Eyshi Rezaei et al., 2015). The reviewer is right that climate also exerts a control on the root zone. An alternative assumption would be to derive a climate dependent root zone, following approaches presented in part of the papers in specific comment #3. However, the adaptation of root zone systems to deal with droughts of certain return period is expected to mainly occur for perennial ecosystems such as grasslands or forests, and lesser for planted annual crops (as also discussed in De Boer-Euser et al. 2016). We further agree with the reviewer about the influence of land cover (crop type) on rooting depth (see also e.g. Fan et al. 2016). We could assume crop specific rooting depth, but such plant specific rooting depths would not match with our more general land use parameterization. In our simulation, we do not consider a variety of different crops, since we carry out a long-term simulation over a spacious area, for which no information about crop rotation is available. We use a parameterization in the model, which takes into account average characteristics (e.g., planting dates, LAI development) derived from a selection of typical crops grown in the area investigated.

We further acknowledge issues regarding our drought definition not being representative. We agree that it is inaccurate to refer to soil moisture drought stress for events with AWC

Simulated soil moisture during the drought of 2003 Absolute vs. Anomaly

**Figure R1**. Simulated soil moisture for an exemplary grid cell. expressed as: percentage left in the potential root zone (upper row) and daily anomaly (lower row).

**How to deal with the assumptions**

Overall, we stress that we prefer the used root zone parameterization and drought identification method within the context of this study for reasons outlined above. However, we acknowledge that we should more carefully discuss the implications of our methods and assumptions, and therefore propose to include.

- A sensitivity analyses on how the parameterization of the root zone and definition of drought affect the derived drought characteristics and controls.
- A more careful discussion on how the derived results change under different assumptions and definitions as well as how derived results might be different from plot scale observations.

Finally, we do not think that we can validate / derive the depth of the root zone of the considered agricultural grid cell with a comparison against streamflow data, given the constraints presented in the reply to major comment #3.

**Proposed sensitivity analyses**

The analyses will be the same as those presented in the manuscript, only with different (1) parameterizations of the root zone and (2) drought definitions and thresholds. We suggest that most of these analyses can be presented in the supplementary material for discussion.

(1) Different parameterizations of the root zone soil

- Soil based AWC as used in the original manuscript.
- Climate based AWC to comply with specific comment #3, although we contemplate whether this should be included in a new version. Calculation procedure (for each grid cell):
  - $\circ\,$  Run the model under optimal conditions (no water stressed reduction in evapotranspiration).
  - Derive maximum soil moisture deficit for each year.
  - o Calculate the maximum deficit that has an expected return period of 10 years.
  - Use the deficit with return period of 10 years as AWC for the final simulations
- Fixed AWC (100 mm & 200 mm). Included to completely remove the effect of the root zone soil characteristics. Two arbitrary values were used to represent both shallow and deep rooting crops.

(2) Different soil moisture drought definitions

- Absolute (%-AWC).
  - Different thresholds to comply with SC-2 (50%, 30%, 10%); emphasizing that 30% is most appropriate (10% is likely to extreme, 50% not really water stress)
- Anomaly based (daily percentiles: rank(SMDOY) / (n+1). Threshold: 20th percentile.
  - Comparing SM in a certain day and year with SM for the same day in other years.

**Results**

In this reply, results are shown for a subset of 100 randomly selected grid cells due to high computational demands. For brevity reasons, only total time in drought is shown. In case of a revised version, these analyses will be carried out for all grid cells and considered characteristics.

**Total time in drought for prominent drought years (1991, 2003, 2015, 2018)**

The ordering of prominent drought years is similar, independent of the used root zone parameterization, drought identification method or threshold (Figs. R2, R3). This would mean that, independent of the used method, we would reach the same conclusion about which drought year was more severe according to total time in drought. However, absolute differences in total time in drought vary substantially, especially among methods. Anomaly based definitions generally result in higher total time below the threshold (Fig. R2). In addition, differences between root zone parameterizations are more obvious for anomaly-based definitions (e.g. Fig. R2g vs. R2h). Obviously, increasing the drought threshold increases the total time in drought (Fig. R3); However, the ordering of major drought years often does not change.

---

## Referee Comment (RC2) · Eric Hunt (Referee) · 30 Aug 2020

Thank you for giving me the opportunity to review this paper. Overall, this paper represented results from a well-research project, is well-written, and will make a very nice addition to the literature once it is revised. The figures are of excellent quality.

The only major issue is the authors terminology of defining drought for soil moisture in absolute terms as opposed to as an anomaly. While opting to look at soil moisture as a % of available water content (putting a current observation in the context of field capacity and wilting point) is highly appropriate, many members of the drought community would take significant issue with saying anything under 30% AWC is drought, if

that occurs more than 20% of the time for a given location and time period. However, what the authors are conveying in the paper is soil moisture stress, or perhaps more correctly- low enough soil moisture to cause significant water stress for vegetation, in the context of drought and flash drought formation. Therefore, I ecommend the authors consider changing the term "SM drought" to "SM stress". This would in no way reduce the importance of the article or the effectiveness of the message. Clearly in years like 2003, 2015, 2018, and 1991, SM drought was appropriate but in other years it may not be, especially for grid points where that is a common occurrence.

Another thing for the authors to consider is to look at the development time and see what percent of cases were more flash drought oriented (e.g., 25-40 days from start to ) vs. a more traditional drought that develops more slowly. That could then be tied to the temperature and precipitation anomalies, in addition to what is already shown in Figure 7.

Finally, please make it more clear in the methods that Socc is only for a particular year and grid point combination. This is implied in the article but more explicitly stating it will help the readers.

Specific comments are as follows: Line 39: List some examples of drought indices and their references L52-54:There are indices (e.g., ESI) that account for both ET and potential ET. L65: As in future climate scenarios or forecasting of soil moisture at S2S? L71-72: Consider re-writing sentence on drought. L82: Below normal precipitation? L131: What does TRAIN stand for? L156: Please clarify the length of spin-up time for the model? Was it truly 1 year (1988) or all 31 years and only 1989-2018 considered in the analysis? If the former, you will need to provide justification for doing so. L177: Elaborate further on why you chose to use FC as opposed to an AWC of say above 0.70.

---

## Author Comment (AC2) · 3 Sep 2020

**Reply to the comments of Eric Hunt**

Thank you for giving me the opportunity to review this paper. Overall, this paper represented results from a well-research project, is well-written, and will make a very nice addition to the literature once it is revised. The figures are of excellent quality.

We thank Eric Hunt for his positive and constructive feedback on our manuscript. Below, we provide a first reply (in blue) to his comments (which are shown in black). A detailed reply, including some complementary analyses related to comments, will be provided after the editor's decision and once the first author is back from parental leave (10.05.2020).

The only major issue is the authors terminology of defining drought for soil moisture in absolute terms as opposed to as an anomaly. While opting to look at soil moisture as a % of available water content (putting a current observation in the context of field capacity and wilting point) is highly appropriate, many members of the drought community would take significant issue with saying anything under 30% AWC is drought, if that occurs more than 20% of the time for a given location and time period. However, what the authors are conveying in the paper is soil moisture stress, or perhaps more correctly- low enough soil moisture to cause significant water stress for vegetation, in the context of drought and flash drought formation. Therefore, I ecommend the authors consider changing the term "SM drought" to "SM stress". This would in no way reduce the importance of the article or the effectiveness of the message. Clearly in years like 2003, 2015, 2018, and 1991, SM drought was appropriate but in other years it may not be, especially for grid points where that is a common occurrence.

We fully agree that using the term "drought" for a phenomenon that might not be below normal contradicts the definition of drought. We gladly take over the suggestion and refer to low enough soil moisture to (likely) cause significant water stress for vegetation, hereafter referred to as "SM stress" for brevity reasons. To comply with the comments of #R1, we will make sure to carefully discuss assumptions related to the used definition. In addition, we can create a Figure complementary to Fig. 8 (i.e., Fig. 8b), to discuss how rare "SM stress" is for a certain DOY, i.e., which percentile value represents 30% AWC. Such figure might provide an informative comparison, as there can be quite a variability in what "below normal" means depending on time of year and location.

Another thing for the authors to consider is to look at the development time and see what percent of cases were more flash drought oriented (e.g., 25-40 days from start to ) vs. a more traditional drought that develops more slowly. That could then be tied to the temperature and precipitation anomalies, in addition to what is already shown in Figure 7.

We will characterize:

- The percentage of droughts that were more flash drought oriented vs. more traditional. This can be connected to Fig. 6b (which shows the distribution of development times).
- The meteorological anomalies during the development of flash droughts vs. slower developing traditional droughts. This will be connected as suggested to Fig. 7.

We will further add a similar kind of analyses with development time starting once SM reached 70% AWC (related to your last specific suggestion).

Finally, please make it more clear in the methods that Socc is only for a particular year and grid point combination. This is implied in the article but more explicitly stating it will help the readers.

We will emphasize in section 2.4 that $S_{occ}$ was derived for each individual grid cell and year.

Specific comments are as follows:

Line 39: List some examples of drought indices and their references

We will add a new sentence that lists some examples of drought indices relevant for this study. This list will include some of the commonly used (standardized) meteorological indices such as the SPI and SPEI as well as some agricultural drought indices such as the Soil Moisture drought Index (SMI) or the Evaporative Stress Index (ESI) (McKee et al., 1993; Samaniego et al., 2012; Vicente-Serrano et al. 2010; Anderson et al. 2007).

L52-54:There are indices (e.g., ESI) that account for both ET and potential ET.

We will introduce indices that include both ET and PET, such as the ESI, in this paragraph.

L65: As in future climate scenarios or forecasting of soil moisture at S2S?

We will clarify that future refers to climate change scenarios.

L71-72: Consider re-writing sentence on drought.

We will rewrite this sentence.

L82: Below normal precipitation?

We will clarify that in this case "below normal anomaly" refers to a below normal anomaly of different hydrometeorological variables.

L131: What does TRAIN stand for?

We will mention that TRAIN stands for TRAnspiration and INterception (evaporation) since these were the most important elements at the time the model was originally developed.

L156: Please clarify the length of spin-up time for the model? Was it truly 1 year (1988) or all 31 years and only 1989-2018 considered in the analysis? If the former, you will need to provide justification for doing so.

The spin-up time of the model was 1 year, i.e., we started the simulations 1 year prior to the considered period. This was mainly done to get the initial condition with regard to snow right at the beginning of simulations. A longer spin-up time was not needed in this case, as within the study region there is no multiyear snow accumulation and the considered agricultural grid cells reach field capacity before the start of the new growing season. Further, a comparison with longer spin-up times (1984-1988) reveals that results are not affected by this.

L177: Elaborate further on why you chose to use FC as opposed to an AWC of say above 0.70.

This is an interesting remark. We use FC as it is a nice absolute quantity, i.e., from here SM started to deplete. However, we could definitely characterize the development starting from e.g. 70% AWC, which would be closer to the transition from no stress towards stress. Such an additional analyses could be integrated in quite a straightforward manner in Fig. 6 and possibly also in a modified Fig. 7 (see reply to second comment).

**References:**

Anderson, M. C., Norman, J. M., Mecikalski, J. R., Otkin, J. A., and Kustas, W. P.: A climatological study of evapotranspiration and moisture stress across the continental U.S. based on thermal remote sensing: 2. Surface moisture climatology, J. Geophys. Res., 112, https://doi.org/10.1029/2006JD007507, 2007.

McKee, T. B., Doesken, N. J., and Kleist, J.: The relationship of drought frequency and duration to time scale, in: 8th Conference on Applied Climatology, 17–22 January, Anaheim, California, American Meteorological Society, Boston, 179–184, 1993.

Samaniego, L., Kumar, R. and Zink, M.: Implications of Parameter Uncertainty on Soil Moisture Drought Analysis in Germany. J. Hydrometeorol., 14, 47–68. https://doi.org/10.1175/jhm-d-12-075.1, 2012.

Vicente-Serrano, S. M., Beguería, S. and López-Moreno, J. I.: A multiscalar drought index sensitive to global warming: The standardized precipitation evapotranspiration index, J. Climate, 23, 1696–1718, https://doi.org/10.1175/2009JCLI2909.1, 2010.

---

## Author Comment (AC3) · 26 Sep 2020

**Reply to the short comments of Chunyu Dong and Zhiyong Liu**

We would like to thank Chunyu Dong and Zhiyong Liu for the provided comments and remarks. Before replying, we would like to mention that Chunyu Dong and Zhiyong Liu are former PhD students of the second author, now working in China.

**Short comments of Chunyu Dong**

The paper by Tijdeman and Menzel investigated the spatial and temporal variability of soil moisture drought in southwestern Germany using a distributed hydrological model. They analyzed the environmental controls on drought development and duration. Some interesting results are presented in the paper. For example, they find that drought stress tends to occur in warmer and drier locations. This raises an important question, i.e. how did and will climate change affect drought occurrence and severity? Some researchers have conducted some attempts to answer this question at a large scale, e.g. Samaniego et al. (2018). I would suggest the authors do some work on. For example, the authors may simulate the drought stress at different warming and precipitation change scenarios, and see what land covers will be affected most. I know that this analysis is already beyond the scope of this paper. The authors do not need to add this analysis in this paper. I have some other comments below, which may be helpful for the authors to improve this manuscript.

We appreciate the suggestions for future work.

L102. Figure 1a. I would suggest clipping the elevation map and only reserve the Baden-Württemberg. Then you may add a small panel at the corner which indicates the location of Baden-Württemberg in Germany.

Thanks for this suggestion. We will modify Figure 1a accordingly.

L169. In this paper, the SM drought threshold is set to 30% of AWC. Then a binary time series of SM drought stress occurrence becomes the basic data of this study. I am thinking it might be helpful to further classify the SM drought to different levels, for example, moderate, severe and extreme SM droughts. In this case, Figure 4 may demonstrates the temporal variations of cell counts for different drought severity. It may provide some information like whether climate warming has increased the drought severity in this region.

We can definitely add this information to the Figures. For example, see Figure R3 in the reply to RC1. This Figure shows that derived drought characteristics are (as expected) depending on the used severity threshold. However, the relative ordering according to severity is often not affected. In our study, we do not study a sufficiently long record to make strong claims about trends in simulated results.

(4) L170-175. SM drought stress occurrence (Socc) was computed in the basis of calendar year in this paper. Normally, most of the soil moisture droughts in Germany happen between spring and autumn. However, was there some winter droughts over 1989-2018, which began at the end of a year and ended in the following year? If yes, these special circumstances may overestimate the drought occurrences in the successive two years. In addition, how did you calculate the development time and duration for these special droughts? I assume these events are very rare in this region.

Simulations reveal that, according to the used drought definition and the selected agricultural grid cells, there were no multiyear droughts. All grid cells were filled above 30% AWC before the start of the next year and filled to field capacity before the start of the next growing season. Thus, winter droughts were not apparent for the considered agricultural grid cells.

(5) L225-227. This sentence is confusing. What does "the latter" refers to? What I see is that drought tends to develop at warmer locations for all prominent drought years but not for all the other years. Please make it more clear.

The sentence was meant as described above, but we agree that the use of "the latter" was vague and we will rephrase it.

(6) L255-258. It would be helpful to add the significance test of the rank correlations in Table 1

We could add significance, but question the value in thuis particular case, because given the large sample size, low correlations will already be significant. We think the relative magnitude and sign of the correlation coefficients provide enough information.

**Short comment of Zhiyong Liu**

It is interesting to read the entitled paper "Controls on the development and persistence of soil moisture drought across Southwestern Germany". The manuscript is well-written and well-organized. I have a few suggestions regarding the method. It seems the authors used the linear correlation and regression models to identify the individual contribution from different controls on SM drought features. However, these two approaches can not differ the co-influencing between the controls (i.e., the soil properties and climate settings). Probably, the partial least squares regression (PLSR) and the partial correlation analysis could be more efficient to identify the individual Kindly see the R function, e.g., plsr and pcor.test in R program. The results based on the PLSR and partial correlation could be different from the current results. The Authors could make some tests based on their sample data. It is only a suggestion.

We will explore the suggested statistical techniques. However, in a revised manuscript, we would weaken our claims about controls, given the remarks of reviewer 1 that these are model based and reliant on assumptions. Therefore, we are hesitant to include additional statistical analyses.

---

## Author Response (AR1)

**Reply to the editor and reviewers indicating changes made to the manuscript**

**Reply to the editor**

**Dear authors,**

as you have seen, the two reviewers provide an excellent list of detailed comments on your manuscript. Although they both appreciate your efforts and agree on the general interest of your study, they nevertheless provide very mixed assessments.

I agree with the main points made by Reviewer #1, that (1) AWC may not be the most suitable choice to quantify root-zone storage characteristics and that (2) insufficient information is given on the implementation and evaluation of the model used here - what were the choices made here and why? How does this influence the interpretation?

I encourage you to address these points and all other comments and incorporate them in a meaningful way in a revised version of your manuscript.

**Best regards, Markus Hrachowitz**

**Dear Markus Hrachowitz,**

Thank you for your evaluation of the reviews of our manuscript. Below, we note how and where we implemented the comments of the reviewers, or specifically state when and why we did not do so. With regard to your main remarks:

- (1) We fully agree that the Srootzone derived from soil parameters might not always agree with the actual amount of water available for crops. However, given the focus of our regional study on agricultural grid cells, we prefer the chosen Srootzone parameterization as our main approach as it considers a lower boundary, i.e., the depth is constrained to a level below which roots are unlikely to develop (Section 2.3, line 206-215). This does not mean that roots of agricultural crops make full use of the entire rootzone, as was pointed out by reviewer #1 and mentioned in the manuscript (Section 4, line 573-574). Therefore, we included additional sensitivity analyses, i.e., doing the same analyses but with a different parameterization of the root zone (description in new Section 2.8). Although the sensitivity analyses will not provide the answer about the most optimal  $S_{\text{rootzone}}$  parameterization across space and time, a topic that we believe would fit better in a separate study with a different scope, it will at least provide some feasible scenarios of Srootzone for the considered agricultural grid cells. Finally, we would like to mention that we acknowledge the idea of a climate-based root zone mentioned by reviewer #1 and discuss this possibility in Section 4 (line 580-589). However, we don't think it is feasible to include this in the current study, given that our focus is on agricultural grid cells with annual crops, whereas a climate based rootzone works with the hypothesis that roots develop (over time) to deal with droughts of certain return periods.
- (2) The implementation and structure of the model, and choices made in the set-up of the model, are now carefully outlined in Section 2.3 (flowchart of the model in new Fig. 2, model description on lines 170-186, modeling assumptions explained together with some critical remarks on why these assumptions differ from reality on line 192-215). The way the model was evaluated is described in new Section 2.4 and results of this evaluation are presented in Section 3.1 and the new supplementary material (Fig. S2-S4). The sensitivity analyses, investigating the impact of changing the root zone storage, as well as the impact of investigating SM drought instead of SM stress, is outlined in section 2.8. The way how this changed the interpretation of the results is now presented (Section 3.3, Fig. S5-S8) and

discussed (Section 4). Finally, we would like to mention here that we prefer a sensitivity analyses using scenarios of rooting depth over some kind of optimization analyses trying to find the  $S_{rootzone}$  that results in the best model performance. This, because such an optimization exercise requires a study with a different scope, i.e., focusing on more forested or grassland catchments within the study area. Further, we can modify / calibrate  $S_{rootzone}$  but will not know whether we get a better match with observed streamflow because we simulate SM in the agricultural grid cells better or because for other reason, e.g., we simulate SM in forest or grassland grid cells better.

Kind regards,

**Erik Tijdeman and Lucas Menzel**

**Reply to the comments of the reviewers.**

We would like to thank the reviewers again for their critical and constructive remarks on our manuscript. Below, we indicate (in blue) how and where we implemented the comments (shown in black) in the revised version of the manuscript. Line numbers, sections and figure numbers refer to the track-changed manuscript (below) or the newly added supplementary material. We mainly focus on highlighting the changes made to the manuscript given that more detailed explanations were already provided in the previous replies to each review.

**Reply to the comments of reviewer #1**

**Major comments.**

The data used for the available water-holding capacity (AWC, i.e., the amount of plant available water in the root zone at field capacity), might not be representative for the actual amount of water available to vegetation at all and could be significantly biased as climate and land cover types are in reality the main controls on root zone storage capacity and not the soil type. This would be fine, however, if we would accept that AWC is simply a soil characteristic, but then the definition of soil moisture stress occuring at 30% AWC might be biased instead.

- We now clearly explain how we derived the AWC of *S*rootzone (Section 2.2 line 127-129, Section 2.3 line 206-212).
- We justify why we used a soil-based definition of *S*rootzone and further acknowledge that this might not be representative of real-world conditions (Section 2.3, line 212-215).
- We then propose a sensitivity analysis in Section 2.8, i.e., what if:
  - We use a soil-based definition of the root zone but constrain the maximum depth to one meter
  - We use a root zone with a fixed AWC of resp. 100 mm and 200 mm, specifically focusing on certain crop type with a low or high amount of water availability.
- With the sensitivity analysis, we present (new Section 3.3, Fig. S5-S8) and discuss (throughout section 4) how:
  - SM stress characteristics change and,
  - Controls on SM stress characteristics change,
  - depending on the used parameterization of the root zone.
- We did not include a climate based Srootzone but discuss this possibility in Section 4 (Line 580-589). The in this paragraph mentioned reasons a climate based Srootzone was not included were:

- Because we do not think this approach works well for the Agricultural grid cells, i.e., the hypothesis behind the climate-based root zone is that it adapts over time to deal with droughts of certain return periods, whereas annual agricultural crops cannot do that.
- It would require an analysis of a different scope to test whether a climate-based root zone results in better modeling results.
- Overall, we still do not know for sure how much water plants have access to across the study region and over the considered period of record, and stress this again in the discussion (Section 4, Line 586-588) and conclusion (Section 5, Line 664-665). That being said, this is a common problem in regional models, yet they are being used for drought assessments. The additional sensitivity to the used root zone parameterization provides insight on how robust or not the derived results are, whereas the sensitivity analyses comparing SM stress and drought events point towards some interesting differences between the two different metrics.

- Conclusions are drawn on AWC being a control of reaching 30% AWC. This is clearly circular reasoning and those findings can hardly be considered surprising.

- We agree that these findings are obvious. Nonetheless, we believe that they are worth showing, also in the context of the use of meteorological proxies of SM drought (Section 4, Line 517-519).
- We think the "unsurprising" findings become more interesting with the newly included sensitivity analyses, especially, the differences between SM drought and SM stress. We now show the AWC has an obvious control on SM stress characteristics, whereas it has little or sometimes even a contrasting control on SM drought characteristics (Section 2.8, 3.3, 4, Fig. S5-S8). This provides some interesting material for discussion on different methods for agricultural drought assessments (Section 4, line 605-620; new Fig. 10 and S10), i.e., SM stress, shows some obvious relationships with the AWC, whereas SM drought does not show the obvious relation with the AWC (but is therefore more robust to uncertainties in the parameterization of the root zone).

This study evaluates the soil moisture within a hydrological model (TRAIN), however, there is no information shown on the setup of the model and whether this model performs well at all based on streamflow or other measurements. This might be shown in the papers that are referred to, but I would find it useful here as well. Neither is it evaluated how crucial information/parameterization affects the results. Does the in- or exclusion of the AWC data vs. a fixed value improve model performance? Is the vegetation water stress formulation in TRAIN really the best and would other parameters lead to worse or better streamflow predictions?

- We now provide a conceptual model overview, which displays all fluxes and stores of the TRAIN model (new Fig. 2) as well as a description how these are derived (Section 2.3, lines 170-186).
- It is worth mentioning that TRAIN has been developed to simulate the water fluxes at the soil-vegetation-atmosphere interface, i.e., it is not a rainfall-runoff model (see Fig. 2). Focus in the model is on the control of evapotranspiration through water availability in the soil. Thus, it necessarily includes a vegetation water stress approach which has been deduced from own research and recent findings documented in the literature. However, we do not know if it is really "the best" if such a best approach really exists.

- Based on the aggregated water fluxes generated by the model, we included a model evaluation using streamflow of 60 catchments with near-natural flow located across the study area (new Fig. S1). This evaluation is described in new section 2.4 and includes:
  - A comparison between the simulated and observed average annual water balance (new Fig. 5a)
  - An assessment of the correlation between annual simulated and observed water balance (Fig. 5b)
  - An assessment of the agreement between SM and river flow (during drought years), under the hypothesis that the drying of SM caused by meteorological dry spells should also be visible for some catchments (Fig. S2)
  - An investigation whether most event flow occurs when simulated SM of the majority grid cells in the catchment exceed field capacity (Fig. S3, S4).

Overall, these results suggest that the TRAIN model provides a reasonable estimate of the simulated fluxes and stores.

- We argue against a calibration of the root zone against streamflow observations in the context of this research, as we do not know whether a possible improvement according to one of the points above relates to a more realistic representation of the root zone for the considered agricultural grid cells. This, because each catchment encompasses a mixture of different land uses (See also reply to major comment #1). Rather, we investigated how the derived drought characteristics change depending on some possible parameterizations of the root zone (described in new section 2.8).

**Specific comments**

**#1: "L38: "Droughts are often defined as a below normal water availability"**

I would have expected some critical reflections on this directly in or directly after this paragraph and not by the end of the introduction."

Critical reflection is now provided directly in this paragraph (Section 1, line 48-50).

**#2: "L75: "which is indicative for low soil moisture levels causing drought stress for plant"**

Given the fact that at this point in the introduction drought has only be described to be defined as an anomaly and not as an absolute measure, low soil moisture levels can occur without having a drought, so the plants in this example just experience water stress and not drought stress."

Changed to soil moisture stress (Section 1, line 93).

**#3:** "L109: "Vectorized soil property data (field capacity and wilting point of the root zone soil) were derived from the BK-50 (scale of 1:50,000) dataset provided by the Federal State Office for Geology Resources and Mining (LGRB, 2019)."**

Is this the available water-holding capacity in the rootzone? Does it include thickness as well as soil type? This is not clear. More importantly: how do you know that plants' roots really access all this water? There have been many studies showing that the root zone storage capacity is not a characteristic of the soil, but mainly that of the climate and the plant (e.g., de Boer-Euser et al., 2016; Fan et al., 2017; Gao et al., 2014; Guswa, 2008; Kleidon, 2004; Nijzink et al., 2016; Speich et al., 2018). Therefore, it should be made clear in the manuscript that AWC is a soil property within a part of the rootzone, but not necessarily a characteristic of the rootzone itself, and may even be completely unrelated to root zone water storage capacity."

More information about the root zone soil data is added to section 2.2 (Line 128) and 2.3 (Line 206-212). This data includes thickness as well as an estimate of the AWC from soil properties. A justification why we make this assumption is added to Section 2.3 (line 212-215), i.e., we argue why we prefer the used assumption in this regional modeling study and mention that it is more often used, but also briefly note why this assumption might be different from reality. We added a sensitivity analyses that investigates the impact of some alternative parameterizations of  $S_{rootzone}$  on the derived results, i.e., constraining the depth or the size of the  $S_{rootzone}$  to differentiate between shallow and deep rooting crops (Section 2.8). We added a note to the discussion about climate-based root zones (Section 4, line 582-586), but argue that this is less applicable in our study, given that we focus on agricultural grid cells. This note will also be added to the dataset that accompanies this paper. Together with some additional remarks in the discussion and conclusion (e.g., Line 587-589, 597-599, 644-646), we sufficiently clarified that the assumed root zone might differ from the actual root zone.

**#4: "L145-146:** "Thus, the root zone soil is not subdivided into different layers but understood as one uniform soil column."

Does it have a specific pre-defined thickness? Was it calibrated on something? This is a crucial parameter, so a more comprehensive description would be useful to the reader."

The way the soil thickness is incorporated is now explained in Section 2.2 and 2.3. The thickness and AWC of  $S_{\text{rootzone}}$  were not calibrated, for reasons mentioned in the reply to major comment #3.

**#5**: "L218-L220: "The latter suggests a stronger influence of root zone soil characteristics, over the influence of the climatological setting, on whether or not SM drought stress developed. SM drought stress was further found to be more likely to develop in soils that have a lower AWC (Fig. 5a), as the likelihood of Socc increases with decreasing AWC.""

Yes, obviously this is the case. The probability of occurrence of SM drought stress (defined as <30% of AWC!) is related to AWC. It's extremely obvious that these variables are related, so it's not surprising at all to find a strong relation, especially as this is an entirely model-determined results. This is clearly circular reasoning and can hardly be considered surprising."

See the reply to major comment #2.

**#6:** "L302-L303: "SM drought stress was generally more likely to develop, and evolved faster and earlier in the year, in shallow root zones with a lower AWC."**

Yes, obviously this is the case as SM drought stress is defined as <30% of AWC! This is again clearly circular reasoning and can hardly be considered surprising."

**See the reply to major comment #2.**

**#7**: "L305-L306: "Results also confirm that AWC of the root zone is an important factor to determine the vulnerability to agricultural drought"**

In your model that is and with a definition where agricultural drought is defined as a percentage of AWC. This conclusion is, therefore, overstated and should be withdrawn in case it cannot be backed up with any observations (crop yields, vegetation observations, etc.) or hard proof that the hydrological model is a reliable descriptor of true states and fluxes."

The conclusion is rephrased and more carefully stated using the additional sensitivity analyses (Section 5, line 639-643). In addition, TRAIN has proven to be reliable on the plot-scale (e.g. studies mentioned in Section 2.3) and the newly included comparison against streamflow observations are reasonable as well (Section 3.1, Fig. 5, S1-S4).

**#8: "L352: "However, roots do not necessarily utilize the water in the entire soil column"**

Exactly! Or they are able to access more water than what you think based on the soil map and model parameterization. There would likely be great differences between forests, grasses and crops and the roots would develop differently under different climates. Therefore, what you define as soil moisture drought stress could be far from reality."

We emphasize that the focus is on agricultural grid cells. (e.g., Abstract, Line 18; Section 2.3, Line 200, Section 2.5, Line 260). We now refer to SM stress instead of SM drought stress (throughout the manuscript), and now discuss why our simulations might differ from reality (Section 2.3, Line 206-215; Section 4, Line 573-589). We further note that this is a common issue with regional drought simulation studies, which more often use soil based definitions of the root zone. This is also (or especially) the case when studying SM drought, i.e., SM might be anomalously low but that does not it has the potential to cause drought impacts (Section 4, line 608-614).

**#9: "L357**: "However, by analyzing a large sample of grid cells, we cover most combinations of root zone characteristics and climatological settings that occur within the study region"

Even if we accept that the rootzone characteristics and climate to be wrongly represented in individual grid cells, you have no basis to claim that the probability distribution function of root zone vs. climate is representative of reality."

We now investigate how probability distribution functions change under different assumptions of the root zone (Section 3.3). We use this, and other reasons, to discuss that probability functions are only valid if the assumptions behind the simulations apply, and that studies with different scopes can have different assumptions (Section 4, Line 597-599).

**Minor comments**

Reviewer #1

L34: "aerial overview" What does this mean? Aerial in the literal sense or as a figure of speech? Perhaps just use overview.

Changed to just overview as suggested (Section 1, Line 43)

L71: "it's"

The word ""it's" is removed (Section 1, Line 85).

The numbers on the side are probably some kind of coordinates, but not defined. Moreover, all text is really small and difficult to read.

We re-projected the maps to the Latitude Longitude coordinate system and further increased the size of all labels (modified Fig. 1, modified Fig. 3).

L114: "watt/m2" Just an example, but notation should be W m-2 (please look at HESS Mathematical requirements)

Changed as suggested throughout the manuscript

L123: "T, Uspeed, RH and RG" Just an example, please avoid acronyms where a single symbol could easily be used and use italic notation for physical quantities (please look at HESS Mathematical requirements)

Throughout the manuscript, we now use italic notation for all physical quantities. We further either removed these two letter abbreviations (as we only used them once or twice) or kept them in case they are more commonly used in literature (e.g. SM, AWC).

Fig. 3. Units missing in the legend.

Units are now added to the legend (modified Fig. 4).

L204-L205: "For ease of notation, we omit the grid cell and year identifiers (i and y) from the variable subscripts in the remainder of this paper." I don't think it was necessary then to introduce i in the first place. Moreover, y is used in the remainder of the manuscript making the statement incorrect.

We believe that it is helpful to introduce i, to emphasize that SM stress characteristics were calculated for each agricultural grid cell i (and year) separately (see also third comment of reviewer #2). We rephrased the sentence where it now notes that grid cell identifiers are omitted and year identifiers are omitted where applicable (Line 301-302).

Fig. 5 and beyond. What is defined here as likelihood should be probability. There is no hypothesis testing or anything that would justify using the term likelihood.

We now refer to probability throughout the manuscript.

L229. "at least once in a year (Socc = 1)". The symbol of at least once is  $\geq$  and not =.

 $S_{occ}$  refers to the binary timeseries, which indicate whether SM stress was reached for at least one day or not. Therefore "=" is correct in this case.

L330: "vegetative stress" Water stress for vegetation

Changed as suggested (Section 4, Line 548).

**Reply to the comments of Eric Hunt (reviewer #2)**

The only major issue is the authors terminology of defining drought for soil moisture in absolute terms as opposed to as an anomaly. While opting to look at soil moisture as a % of available water content (putting a current observation in the context of field capacity and wilting point) is highly appropriate, many members of the drought community would take significant issue with saying anything under 30% AWC is drought, if that occurs more than 20% of the time for a given location and time period. However, what the authors are conveying in the paper is soil moisture stress, or perhaps more correctly- low enough soil moisture to cause significant water stress for vegetation, in the context of drought and flash drought formation. Therefore, I ecommend the authors consider changing the term "SM drought" to "SM stress". This would in no way reduce the importance of the article or the effectiveness of the message. Clearly in years like 2003, 2015, 2018, and 1991, SM drought was appropriate but in other years it may not be, especially for grid points where that is a common occurrence.

- We changed the term "SM drought stress" to "SM stress" throughout the manuscript.
- We added a Figure to the discussion to show how uncommon (or anomalous) SM stress is (Fig, 10) and discuss that SM stress is often still an anomalously low event that develops during periods with below normal precipitation (Section 4, line 609-620).

Another thing for the authors to consider is to look at the development time and see what percent of cases were more flash drought oriented (e.g., 25-40 days from start to ) vs. a more traditional drought

that develops more slowly. That could then be tied to the temperature and precipitation anomalies, in addition to what is already shown in Figure 7.

- We changed former Fig. 7 in the following way:
  - Split al SM stress events in quickly developing, flash drought-oriented events (<30 days) and slower, more traditional developing events (modified Fig. 9a).
  - Split al SM stress events in shorter (<30 days) and longer events (modified Fig. 9b).

Interestingly, the meteorological conditions during the more flash-drought oriented events ten to be more extreme (relatively lower precipitation and higher temperature and evapotranspiration).

- We mention the percentages of SM stress events belonging to each category (short / long) in the caption of the Figure.

Finally, please make it more clear in the methods that Socc is only for a particular year and grid point combination. This is implied in the article but more explicitly stating it will help the readers.

**We added an extra clarifying remark (Section 2.5, Line 265-266)**

Line 39: List some examples of drought indices and their references

We replaced the more general reference to Lloyd-Hughes (2014) with a short list of some commonly used (standardized) meteorological indices relevant for our study (Section 1, Line 52-53).

**L52-54:There are indices (e.g., ESI) that account for both ET and potential ET.**

We introduced indices such as the ESI in this paragraph (Section 1, Line 61-63)

**L65: As in future climate scenarios or forecasting of soil moisture at S2S?**

We clarified that the sentence refers to climate change scenarios (Section 1, line 79).

**L71-72: Consider re-writing sentence on drought.**

We rephrased the part of the sentence that mentioned that there is a common consensus on the slowly developing nature of drought (Section 1, line 86).

**L82: Below normal precipitation?**

We added a note that below normal refers to hydrometeorological variables (Section 1, line 96-97).

**L131: What does TRAIN stand for?**

We mention that TRAIN is an abbreviation of TRAnspiration and INterception (Section 2.3, line 150).

L156: Please clarify the length of spin-up time for the model? Was it truly 1 year (1988) or all 31 years and only 1989-2018 considered in the analysis? If the former, you will need to provide justification for doing so.

**We now justify having only one warmup year in Section 2.3 (line 226-227).**

**Elaborate further on why you chose to use FC as opposed to an AWC of say above 0.70.**

We added a Figure to the supplementary material where we show how development time changes if we change the starting point to different AWC values (Fig. S9) and discuss how much faster development time becomes if the initial conditions are different (Section 4, line 484-486).

**Reply to the comment of Zhiyong Liu**

It seems the authors used the linear correlation and regression models to identify the individual contribution from different controls on SM drought features. However, these two approaches can not differ the co-influencing between the controls (i.e., the soil properties and climate settings). Probably, the partial least squares regression (PLSR) and the partial correlation analysis could be more efficient to identify the individual. Kindly see the R function, e.g., plsr and pcor.test in R program. The results based on the PLSR and partial correlation could be different from the current results. The Authors could make some tests based on their sample data. It is only a suggestion.

In the end, we did not included partial correlation, but rather included a sensitivity analysis to investigate what happens to the correlation when changing or constraining the AWC of the root zone (section 3.3, Fig. S7).

**Reply to the comments of Chunyu Dong**

Figure 1a. I would suggest clipping the elevation map and only reserve the Baden-Württemberg. Then you may add a small panel at the corner which indicates the location of Baden-Württemberg in Germany

**Applied as suggested in modified Figure 1.**

In this paper, the SM drought threshold is set to 30% of AWC. Then a binary time series of SM drought stress occurrence becomes the basic data of this study. I am thinking it might be helpful to further classify the SM drought to different levels, for example, moderate, severe and extreme SM droughts. In this case, Figure 4 may demonstrates the temporal variations of cell counts for different drought severity. It may provide some information like whether climate warming has increased the drought severity in this region-

In the end, we decided not to include the additional thresholds in the manuscript. The sensitivity analyses already added a significant amount of material and adding additional thresholds would mean a threefold increase in this amount.

SM drought stress occurrence (Socc) was computed in the basis of calendar year in this paper. Normally, most of the soil moisture droughts in Germany happen between spring and autumn. However, was there some winter droughts over 1989-2018, which began at the end of a year and ended in the following year? If yes, these special circumstances may overestimate the drought occurrences in the successive two years. In addition, how did you calculate the development time and duration for these special droughts? I assume these events are very rare in this region.

**We mention that multiyear SM stress events did not occur in the study region (Line 503-507).**

This sentence is confusing. What does "the latter" refers to? What I see is that drought tends to develop at warmer locations for all prominent drought years but not for all the other years. Please make it more clear.

We rephrased this sentence as indicated in the comment and removed the latter (Line 363-364).

**(L255-258. It would be helpful to add the significance test of the rank correlations in Table 1**

We did not include significance, as it distracts from the main message, i.e., the sign and magnitude of the correlation, and how it changes depending on the used root zone parameterization method or SM stress / drought identification method, is most important.

[revised manuscript text omitted]

---

## Referee Report (RR1)

**Review of the revision by Tijdeman and Menzel: Controls on the development and persistence of soil moisture drought across Southwestern Germany**

**General comments**

This manuscript investigates the role of soil characteristics in the root zone and climate properties in determining the probability of occurrence and characteristics of agricultural drought. Although I was very critical towards the initial submission and even recommended rejection with a resubmission, I would like to complement the authors with the way the managed to improve this manuscript. In my opinion only a few issues remain.

- The authors write in their reply that their main focus is on agriculture, hence justifying a purely soil-based (and not climate-based) metric for $S_{rootzone}$. Yet, I do think that the focus on agricultural regions should be mentioned more explicitly at the end of the introduction where they introduce their objectives (end of Section 1), in the description of the study region (Section 2.1), in the description of their modelling approach (Section 2.3) and in the conclusions (Section 5). Also, it should be made clear why some areas that hardly have any agriculture were not left out of the analysis.

- I am happy to see model performance analysis of the TRAIN model, yet, I think this can be taken some steps further in order to rule out the possibility that the conclusions about the importance of AWC are based on systematic errors in estimating the available water to plants (i.e., AWC). Some suggestions:
    - In Figure 5 it would be interesting to see whether there is a relationship between over- or underestimation of the observed flow with AWC. The authors could, for example, include a color scale with AWC and give each dot a color corresponding to the average AWC in that catchment to show, hopefully for the authors, that AWC is not systematically associated with an over- or underestimation.
    - The same analysis as Figure 5 could also be performed on monthly basis. In case this just confirms Fig. 5 just as supplement, but in case significant problems occur, the authors might need to reconsider the fact that AWC in the TRAIN model is not calibrated.
    - Include a figure that answers the question: What is the performance of TRAIN in specific drought years and does or does that not relate to AWC?

**Technical corrections**

Fig. 7. One of the y-axes still has 'likelihood' instead of probability. Please correct.

---

## Author Response (AR2)

**Reply to the editor:**

Dear authors,

thank you very much for submitting a revised version of your work. I have now received the evaluation of one reviewer. The reviewer commends the efforts you invested to address his/her previous comments. I agree with that and think that the manuscript has strongly improved. The reviewer provided a few additional suggestions, which I strongly encourage you to address in a second revision.

I am looking forward to receiving the revised manuscript soon.

best regards,
Markus Hrachowitz

Dear editor,

Thank you very much for your second evaluation of our manuscript. We incorporated the remaining remarks of reviewer #1 in a new version as outlined below.

(1) We placed more emphasis that our focus is on agricultural regions
(2) We added additional model evaluation and show, amongst other things, that there are no systematic biases related to the available water-holding capacity of the root zone.

Kind regards,

Erik Tijdeman and Lucas Menzel

**Reply to the comments of reviewer #1:**

**General comments**

"This manuscript investigates the role of soil characteristics in the root zone and climate properties in determining the probability of occurrence and characteristics of agricultural drought. Although I was very critical towards the initial submission and even recommended rejection with a resubmission, I would like to complement the authors with the way the managed to improve this manuscript. In my opinion only a few issues remain."

We would like to thank the reviewer again for the helpful and critical remarks during two rounds of review. They challenged us to significantly improve the previous versions of the manuscript. Below, we reply (in blue), to the remaining remarks of the reviewer (which are shown in black). The line numbers refer to the track-changed version of the manuscript.

- "The authors write in their reply that their main focus is on agriculture, hence justifying a purely soil-based (and not climate-based) metric for $S_{rootzone}$. Yet, I do think that the focus on agricultural regions should be mentioned more explicitly at the end of the introduction where they introduce their objectives (end of Section 1), in the description of the study region (Section 2.1), in the description of their modelling approach (Section 2.3) and in the conclusions (Section 5). Also, it should be made clear why some areas that hardly have any agriculture were not left out of the analysis. "

- We emphasized our focus on agricultural grid cells at the points suggested: Introduction (Line 98), Section 2.1 (Line 114) and Conclusion (Line 588). It was already stated in Section 2.3 (line 178). The considered agricultural grid cells all have agriculture as their major land use class. These grid cells include areas known for their agricultural production (e.g. Rhine Valley) but also exist in some regions that are not known for their agriculture e.g. some more-local agricultural activities.

- "I am happy to see model performance analysis of the TRAIN model, yet, I think this can be taken some steps further in order to rule out the possibility that the conclusions about the importance of AWC are based on systematic errors in estimating the available water to plants (i.e., AWC). Some suggestions: "

- We evaluated the model based on the suggested evaluation criteria.

   o "In Figure 5 it would be interesting to see whether there is a relationship between over- or underestimation of the observed flow with AWC. The authors could, for example, include a color scale with AWC and give each dot a color corresponding to the average AWC in that catchment to show, hopefully for the authors, that AWC is not systematically associated with an over- or underestimation."

- We changed Figure 5a according to the suggestion. We noted that there are no systematic biases associated with the AWC (Section 3.1, Line 292-293). Thanks for the idea on how we could include this information in the Figure.

   o "The same analysis as Figure 5 could also be performed on monthly basis. In case this just confirms Fig. 5 just as supplement, but in case significant problems occur, the authors might need to reconsider the fact that AWC in the TRAIN model is not calibrated."

- We evaluated the monthly performance of TRAIN. However, as outlined in the revised manuscript, TRAIN does not directly simulate river flow but rather percolation and runoff (Section 2.3, Fig. 2). The delay in groundwater response to streamflow in not considered in the percolation signal. To evaluate monthly model performance, we used the following approach to consider differences in catchment response (inspired from Barker et al. 2016; described in Section 2.4, Line 226-232).

      (1) Accumulate the sum of catchment average percolation and runoff over n-month periods (1-12 months), i.e., percolation in the current month, percolation in the current and previous month etc. (similar to the accumulation of the Standardized Precipitation Index).

      (2) Correlate all accumulated time series with monthly streamflow for each catchment and calendar month

      (3) Identify the accumulation period with the maximum correlation for each catchment and calendar month.

   Given that the found maximum correlation coefficients are of the same magnitude as Fig. 5b, the corresponding Figure is added to the supplementary material (Fig. S2), as suggested, and described in Section 3.1 (line 292-293).

o   "Include a figure that answers the question: What is the performance of TRAIN in specific drought years and does or does that not relate to AWC? "

- We compared for prominent drought years 2003 and 2018 observed streamflow anomalies with anomalies in the sum of simulated percolation and runoff accumulated over the n-month period with the strongest correlation with observed streamflow (Fig. S3). Their anomaly distributions across all catchments were similar (Fig. S3, a-d). The differences between percolation- and streamflow- percentile time series were generally small and unsystematic (Fig. S3, g-h). Finally, biases were not related to the catchment average AWC of the root zone (Figure R1).  Because we now already show that biases are not related to the AWC of the root zone (Fig. 5a), we suggest not to include this Figure. Finally, Fig. S2 in the previous version, showing that periods with below normal river flow coincide with periods of below normal soil moisture, has been integrated in Fig. S3 of the current supplementary material.

[Figure]

**Figure R1.** Difference between observed streamflow percentiles and percentile of the sum of simulated percolation and runoff accumulated over the n-month period that has the strongest correlation with streamflow. Shown for the growing season of prominent drought years 2003 and 2018 for all catchments, i.e. all values shown in Fig. S3g-h, grouped by catchment average AWC.

**Technical corrections**

"Fig. 7. One of the y-axes still has 'likelihood' instead of probability. Please correct."

Corrected as suggested.

**Reference:**

Barker, L. J., Hannaford, J., Chiverton, A., and Svensson, C.: From meteorological to hydrological drought using standardised indicators, Hydrol. Earth Syst. Sci., 20, 2483–2505, https://doi.org/10.5194/hess-20-2483-2016, 2016.